# Diffusion Transformers for Tabular Data Time Series Generation

**Fabrizio Garuti**[1,2,†], **Enver Sangineto**[2,*], **Simone Luetto**[3], **Lorenzo Forni**[1] **& Rita Cucchiara**[2]
1 Prometeia Associazione, Bologna, Italy, 2 University of Modena and Reggio Emilia, Italy,
3 Prometeia SpA, Bologna, Italy.
†fabrizio.garuti@prometeia.com, *enver.sangineto@unimore.it

## Abstract

Tabular data generation has recently attracted a growing interest due to its different application scenarios. However, generating *time series* of tabular data, where each element of the series depends on the others, remains a largely unexplored domain. This gap is probably due to the difficulty of jointly solving different problems, the main of which are the heterogeneity of tabular data (a problem common to non-time-dependent approaches) and the variable length of a time series. In this paper, we propose a Diffusion Transformers (DiTs) based approach for tabular data series generation. Inspired by the recent success of DiTs in image and video generation, we extend this framework to deal with heterogeneous data and variable-length sequences. Using extensive experiments on six datasets, we show that the proposed approach outperforms previous work by a large margin.

## 1 Introduction

Time series of tabular data are time-dependent sequences of tabular rows, where each row is usually composed of a set of both numerical and categorical fields. Tabular data time series are widespread in many real-life applications, and they can represent, e.g., the temporal sequence of the financial transactions of a given bank user or the clinical data of a patient during her hospitalization. Generating time series of tabular data is particularly important due to the limited availability of public datasets in this domain. For instance, despite private banks usually own huge datasets of financial transactions of their clients, they rarely make these data public. A generator trained on a real dataset can synthesize new data without violating privacy and legal constraints. Besides privacy preservation (Jordon et al., 2018; Abdelhameed et al., 2018; Assefa et al., 2020; Efimov et al., 2020; Hernandez et al., 2022; Qian et al., 2023), other common applications of (non necessarily time dependent) tabular data generation include: imputing missing values (Zheng & Charoenphakdee, 2022; Gulati & Roysdon, 2023; Borisov et al., 2023; Zhang et al., 2024), data augmentation and class balancing of real training datasets (Che et al., 2017; Choi et al., 2017; Xu et al., 2019; Kim et al., 2022; Rizzato et al., 2023; Fonseca & Bacao, 2023), and promoting fairness (van Breugel et al., 2021). However, despite the impressive success of generative AI methods for images, videos or text, generating tabular data is still a challenging task due to different reasons. The first problem is the lack of huge unsupervised or weakly supervised datasets for training. For instance, Stable Diffusion (Rombach et al., 2022) was trained on LAION, a weakly supervised dataset of 400 million image-text pairs crawled from the web, while tabular data generation models need to be trained on datasets which are several orders of magnitude smaller. The second reason which makes tabular data generation difficult is the heterogeneity of their input space. In fact, while, for instance, images are composed of numerical pixel values, with a high correlation between adjacent pixels, and textual sentences are sequences of categorical words, tabular data usually contain both numerical and categorical fields (e.g., the transaction amount and the transaction type). Thus, a generative model for tabular data must jointly synthesize both numerical and categorical values, and its training must handle this inhomogeneity (Sec. 2). In case of time series of tabular data, which is the topic addressed in this paper, the additional temporal dimension further increases this difficulty introducing the necessity to model the statistical dependencies among the tabular rows the series is composed of, as well as the need to deal with a variable-length input.

TabGPT (Padhi et al., 2021) and REaLTabFormer (Solatorio & Dupriez, 2023) are among the very few generative deep learning methods for time series of tabular data with heterogeneous field values, and they are both based on a Transformer (Vaswani et al., 2017) trained autoregressively to predict the next input token. Specifically, both methods convert the numerical fields into categorical features, creating field-specific token vocabularies. A time series is then represented as a sequence of concatenated tokens, which are predicted by the network autoregressively. Despite this strategy is effective, the main drawback is the limited diversity of the generated sequences. In fact, an unconditional generation of a time series starts with a specific "start-of-sequence" token and proceeds by selecting the next token using the posterior on the vocabulary computed by the network. However, even when a non-deterministic sampling strategy is used (e.g., randomly drawing from the posterior on the next token), an autoregressive (AR) network tends to generate similar patterns when it is not conditioned on a specific user query. For instance, there is no unconditional sampling mechanism in TabGPT, since it can only *predict* the future evolution of an initial real sequence provided as query.

In this paper, we follow a different direction and we use a Diffusion Model (DM) based paradigm, which is known to be particularly effective in covering multi-modal data distributions (Ho et al., 2020). DMs have recently been applied to the generation of *single-row* tabular data (Kim et al., 2022; Kotelnikov et al., 2023; Lee et al., 2023; Kim et al., 2023; Zhang et al., 2024), empirically showing their superiority both in terms of diversity and realism with respect to other generation paradigms (Zhang et al., 2024). However, none of the existing DM based single-row generation approaches can directly be used to synthesize tabular data *time series*. Indeed, these methods assume relatively short, *fixed-length* input sequences and use denoising networks based on MultiLayer Perceptrons (MLPs). To solve these problems, we propose to use a Diffusion Transformer (DiT) (Peebles & Xie, 2023), which enjoys the benefits of both worlds: it is a Transformer, which can naturally deal with sequences, and it is a DM, which can generate data with a high diversity and realism.

We call our method *TabDiT* (Tabular Diffusion Transformer) and we follow the Latent Diffusion Model (LDM) (Rombach et al., 2022; Peebles & Xie, 2023) framework (Sec. 2), where DMs are trained on the latent space of a pre-trained and frozen variational autoencoder (VAE (Kingma & Welling, 2014)). Very recently, TabSyn (Zhang et al., 2024) exploits a similar LDM paradigm for single-row tabular data generation (Sec. 2). In TabSyn, both categorical and numerical field values are represented as token embeddings, which are fed to a Transformer based VAE. The latent space of this VAE is used as the input space of an MLP-based denoising network. However, TabSyn cannot be used for tabular data time series generation because both its VAE and its denoising network cannot deal with long, variable-length sequences. To solve these issues, we propose to **decompose the representation problem** by simplifying the VAE latent space and using the denoising network to combine independent latent representations. Specifically, since time series training datasets are usually too small to train a VAE to learn complex dynamics with variable length, we split each time series in a set of independent tabular rows which are separately compressed by our VAE. In this way, the variational training is simplified and the time series chunking acts as data augmentation. Thus, differently from previous LDM work, our VAE latent space *does not* holistically represent a domain sample (i.e., a time series), but only its components. Modeling the *combination* of a sequence of embedding vectors in this latent space is delegated to the Transformer denoising network, which is responsible of learning the time-dependent distribution of the time series. Moreover, differently from DM-based single tabular row generation methods, we propose an **AR VAE decoder**, in which the generation of the next field value depends on the previously generated values of the same row, which improves the overall consistency when compared with a standard parallel VAE decoder. Furthermore, since the representation of heterogeneous tabular fields is still a largely unsolved problem, we propose a **variable-range decimal representation** of the numerical field values, based on a sequence of digits preceded by a *magnitude order prefix*. This representation is a trade-off between a lossless coding and the length of the sequence (Sec. 4). Finally, to generate variable-length time series, we follow FiT (Lu et al., 2024), a very recent LDM which deals with variable-resolution image generation by padding the sequence of embedding vectors fed to the DiT. However, at inference time, the image resolution is sampled in FiT independently of the noise vectors fed to the DiT, which is a reasonable choice since the resolution of an image is largely independent of its content. Conversely, a time series of, e.g., financial transactions conditioned on the attributes of a specific bank client, can be longer or shorter depending on the client and how they use their bank account. For this reason, we use the denoising network to also **predict the generated sequence length**, jointly with its content, and we do so by forcing our DiT to explicitly generate the padding vectors defining the end of the time series.

Due to the lack of a unified evaluation protocol for tabular data time series generation, we collect different public datasets and we propose an evaluation metric which extends single-row metrics to the time-series domain. For the specific case of unconditional generation, TabDiT is, to the best of our knowledge, the first deep learning method for unconditional generation of heterogeneous tabular data time series. In this case, we compare TabDiT with a strong AR baseline, which we implemented by merging the (discriminative) hierarchical architecture proposed in (Padhi et al., 2021; Luetto et al., 2023), with some of the architectural solutions we propose in this paper. In all the experiments, TabDiT significantly outperforms all the compared baselines, usually by a large margin. In summary, our contributions are the following.

- We propose a DiT-like approach for tabular data time series generation, in which a Transformer denoising network combines latent embeddings of a non-holistic VAE.
- We propose an AR VAE decoder to model intra-row statistical dependencies and a variable-range decimal representation of the numerical field values.
- We propose to generate variable-length time series by explicit padding prediction.
- We propose a metric for the evaluation of tabular data time series generation, and we empirically validate TabDiT using different public datasets, showing the superiority of TabDiT with respect to the state of the art.
- TabDiT can be used in both a conditional and an unconditional scenario, and it is the first deep learning method for unconditional heterogeneous tabular data time series generation.

## 2 RELATED WORK

**Diffusion Models.** DMs have been popularized by Ho et al. (2020); Dhariwal & Nichol (2021), who showed that they can beat GANs (Goodfellow et al., 2014) in generating images with a higher realism and diversity, reducing the mode-collapse problems of GANs. Stable Diffusion (Rombach et al., 2022) first introduced the LDM paradigm (Sec. 1), where training is split in two phases. In the first stage, a VAE is used to compress the input image. In the second stage, a DM is trained on the VAE latent space. DiT (Peebles & Xie, 2023) adopts this paradigm and shows that the U-Net architecture (Ronneberger et al., 2015) used in Stable Diffusion is not crucial and it can be replaced by a Transformer. Since a U-Net is based on image-specific inductive biases (implemented with convolutional layers), we adopt DiT as our basic framework. Very recently, FiT (Lu et al., 2024) extended DiT to deal with variable-resolution images, by padding the embedding sequence up to a maximum length. We adopt a similar solution to generate variable-length time series. However, in FiT the padding vectors are masked in the attention layers and ignored both at training and at inference time, and the image resolution is sampled independently of the noise vectors. In contrast, we force our DiT to explicitly generate padding vectors, in this way directly deciding about the length of the specific sequence it is generating.

**Single-row tabular data generation.** Early methods for generating *single-row* tabular data include CTGAN (Xu et al., 2019) and TableGAN (Park et al., 2018), based on GANs, and TVAE (Xu et al., 2019), based on VAEs. VAEs have also been employed more recently in other works, such as, for instance, GOGGLE (Liu et al., 2023), where they are used jointly with Graph Neural Networks. TabMT (Gulati & Roysdon, 2023) adopts a Transformer with bidirectional attention to model statistical dependencies among different fields of a row. In GReaT (Borisov et al., 2023), a tabular row is represented using natural language and a Large Language Model (LLM) is used to *conditionally* generate new rows. However, the authors do not provide any mechanism for fully-unconditional sampling. Moreover, the LLM maximum input length limits the dataset size which can be used (Zhang et al., 2024). Finally, tabular field names and values are frequently based on jargon terms and dataset-specific abbreviations, which are usually out-of-distribution for an LLM (Narayan et al., 2022; Luetto et al., 2023).

Most of the recent literature on single-row tabular data generation has adopted a DM paradigm (Kim et al., 2022; Kotelnikov et al., 2023; Lee et al., 2023; Kim et al., 2023; Zhang et al., 2024), and different works mainly differ by the way in which they deal with numerical and categorical features. For instance, both TabDDPM (Kotelnikov et al., 2023) and CoDi (Lee et al., 2023) use a standard Gaussian diffusion process (Ho et al., 2020) for the numerical features and a multinomial diffusion process (Hoogeboom et al., 2021) for the categorical ones. Specifically, CoDi uses a

categorical and a numerical specific denoising network which are conditioned from each other. In STaSy (Kim et al., 2023), both numerical and categorical features are treated as numerical and a self-paced learning strategy (Kumar et al., 2010) is proposed for training. Different from previous work, we propose a variable-range decimal representation that converts numerical features into a sequence of categorical values and allows both categorical and numerical features to be uniformly represented as a sequence of tokens. TabSyn (Zhang et al., 2024) is the work which is the closest to our paper because it is also based on an LDM paradigm (Sec. 1). However, TabSyn cannot be used for *time series* of tabular data because both its VAE and its MLP-based denoising network cannot deal with variable length sequences. For this reason, in this paper we propose a non-holistic VAE, which compresses only a chunk of the input sample (i.e., an individual row rather than an entire time series). In this way, most of the representation complexity is placed on our DiT-based denoising network, which is responsible of generating temporally coherent sequences by combining VAE compressed embedding vectors. Finally, differently from TabSyn and other DM-based tabular data generation methods (Kim et al., 2022; Kotelnikov et al., 2023; Lee et al., 2023; Kim et al., 2023), we use an AR VAE decoder and we explicitly predict the time series length.

**Tabular data sequence generation.** SDV (Patki et al., 2016) is a non-deep-learning method based on Gaussian Copulas to model inter-field dependencies in tabular data. TabGPT (Padhi et al., 2021) (Sec. 1) is a *forecasting* model, which can complete a real time series but it lacks a sampling mechanism for generation from scratch. Moreover, it is necessary to train a specific model for each bank client. In contrast, REaLTabFormer (Solatorio & Dupriez, 2023) has a similar AR Transformer architecture but it focuses on *conditional* generation, in which a time series is generated depending, e.g., on the attributes of a specific bank client, described by a "parent table" (Sec. 3). In REaLTab-Former, numerical values are represented as a fixed sequence of digits. We also represent numerical values as a sequence of digits, but our representation depends on the magnitude order of the specific value, which results in shorter sequences and a reduced decoding error (Sec. 4.1). Differently from (Padhi et al., 2021; Solatorio & Dupriez, 2023), our proposal is based on a DM, and it can be used for both conditional and unconditional generation tasks.

## 3 PRELIMINARIES

**Diffusion Transformer.** The LDM paradigm (Rombach et al., 2022) is based on two separated training stages: the first using a VAE and the second using a Gaussian DM. The main goal of the VAE is to compress the initial input space. Specifically, in DiT (Peebles & Xie, 2023), an image $x$ is compressed into a smaller spatial representation using the VAE encoder $z = E(x)$. The VAE latent representation $z$ is then "patchified" into a sequence $\boldsymbol{s} = [\boldsymbol{z}_1, ..., \boldsymbol{z}_k]$. Note that grouping "pixels" of $z$ into patches is a procedure *external* to the VAE, which holistically compresses the entire image. Then, random noise is added to $\boldsymbol{s}$ and its corrupted version is fed to a Transformer-based denoising network (DiT) which is trained to reverse the diffusion process. More specifically, given a sample $\boldsymbol{s}_0$ extracted from the real data distribution (defined on the VAE latent space, $\boldsymbol{s}_0 \sim q(\boldsymbol{s}_0)$), and a prefixed noise schedule ($\bar{\alpha}_1, ..., \bar{\alpha}_T$), the DM iteratively adds Gaussian noise for $T$ diffusion steps: $q(\boldsymbol{s}_t|\boldsymbol{s}_0) = \mathcal{N}(\boldsymbol{s}_t; \sqrt{\bar{\alpha}_t}\boldsymbol{s}_0, (1 - \bar{\alpha}_t)\boldsymbol{I})$, $t = 1, ..., T$. Using the reparametrization trick, $\boldsymbol{s}_t$ can be obtained by: $\boldsymbol{s}_t = \sqrt{\bar{\alpha}_t}\boldsymbol{s}_0 + \sqrt{1 - \bar{\alpha}_t}\boldsymbol{\epsilon}_t$, where $\boldsymbol{\epsilon}_t \sim \mathcal{N}(\boldsymbol{0}, \boldsymbol{I})$ is a noise vector. A denoising network is trained to invert this process by learning the reverse process: $p_{\boldsymbol{\theta}}(\boldsymbol{s}_{t-1}|\boldsymbol{s}_t) = \mathcal{N}(\boldsymbol{s}_{t-1}; \mu_{\boldsymbol{\theta}}(\boldsymbol{s}_t, t), \sigma_{\boldsymbol{\theta}}(\boldsymbol{s}_t, t)\boldsymbol{I})$. Training is based on minimizing the variational lower bound (VLB) (Kingma & Welling, 2014), which can be simplified to the following loss function (Ho et al., 2020):

$$L(\boldsymbol{\theta})^{Simple} = \mathbb{E}_{\boldsymbol{s}_0 \sim q(\boldsymbol{s}_0), \boldsymbol{\epsilon} \sim \mathcal{N}(\boldsymbol{0}, \boldsymbol{I}), t \sim \mathbb{U}(\{1, ..., T\})} \left[ ||\boldsymbol{\epsilon}_t - \boldsymbol{\epsilon}_{\boldsymbol{\theta}}(\boldsymbol{s}_t, t)||_2^2 \right], \tag{1}$$

where $\mu_{\boldsymbol{\theta}}(\cdot)$ is reparametrized into a noise prediction network $\boldsymbol{\epsilon}_{\boldsymbol{\theta}}(\cdot)$. Following (Nichol & Dhariwal, 2021), DiT predicts also the noise variance ($\sigma_{\boldsymbol{\theta}}(\cdot)$), which is used to compute the KL-divergence of the VLB in closed form. However, for simplicity, we do not predict the noise variance.

At inference time, a latent $z_T$ is sampled and fed to the DiT network, which follows the reverse-process sampling chain for $T$ steps until $z_0$ is generated. Finally, the VAE decoder $D(\cdot)$ decodes $z_0$ into a synthetic image. For *condional* generation, DiT encodes the conditional information (e.g., the desired class label of the image) using a small MLP. The latter regresses, for each DiT block, the parameters of the adaptive layer norm ($\boldsymbol{\beta}, \boldsymbol{\gamma}$) and the scaling parameters ($\boldsymbol{\alpha}$) used in the residual connections. In (Peebles & Xie, 2023), this is called *adaLN-Zero block* conditioning and it is used also to represent the timestep $t$.

**Problem formulation.** Tabular data are characterized by a set of *field names* (or *attributes*) $A = \{a_1, ..., a_k\}$, where each $a_j \in A$ is either a categorical or a numerical attribute. A tabular row $\boldsymbol{r} = [v_1, ..., v_k]$ is a sequence of $k$ *field values*, one per attribute. If $a_j$ is a numerical attribute, then $v_j \in \mathbb{R}$, otherwise $v_j \in V_j$, where $V_j$ is an attribute-specific vocabulary. A time series is a variable-length, time-dependent sequence of rows $\boldsymbol{x} = [\boldsymbol{r}_1, ..., \boldsymbol{r}_\tau]$. Given a training set $X = \{\boldsymbol{x}_1, ..., \boldsymbol{x}_N\}$ empirically representing the real data distribution $q(\boldsymbol{x})$, in *unconditional* tabular data time series generation the goal is to train a generator which can synthesize time series following $q(\boldsymbol{x})$. Moreover, inspired by (Solatorio & Dupriez, 2023), for the *conditional* generation case, we assume to have a "parent" table $P$ associated with the elements in $X$. A tabular row $\boldsymbol{u} = [w_1, ..., w_h]$ in $P$ is associated with a sequence $\boldsymbol{x}_i \in X$ and it describes some characteristics that affect the nature of $\boldsymbol{x}_i$. For instance, if $\boldsymbol{x}_i \in X$ is the history of the transactions of the i-th bank client, the corresponding row in $P$ could describe the attributes (e.g., the age, the gender, etc.) of this client. In a conditional generation task, given $\boldsymbol{u}$, the goal is to generate a synthetic time series according to $q(\boldsymbol{x}|\boldsymbol{u})$.

## 4 METHOD

In this section we present our approach, starting from an unconditional generation scenario, which we will later extend to the conditional case. Our first goal is to simplify the VAE latent space: since the time series have a variable length and a complex dynamics, rather than representing their distribution using a variational approach (Kingma & Welling, 2014), we use our VAE to separately compress only individual *tabular rows*, and then we combine multiple, independent latent representations of rows using the DiT-based DM. In more detail, our VAE encoder $\mathcal{E}_{\boldsymbol{\phi}}$ represents a tabular row $\boldsymbol{r}$ with a latent vector $\boldsymbol{z} = \mathcal{E}_{\boldsymbol{\phi}}(\boldsymbol{r})$, $\boldsymbol{z} \in \mathcal{Z}$, which is decoded using the VAE decoder $\mathcal{D}_{\boldsymbol{\varphi}}$. The parameters $\boldsymbol{\Phi} = [\boldsymbol{\phi}, \boldsymbol{\varphi}]$ of $\mathcal{E}_{\boldsymbol{\phi}}$ and $\mathcal{D}_{\boldsymbol{\varphi}}$ are trained with a convex combination of a reconstruction loss and a KL-divergence (Kingma & Welling, 2014). The training set is $R = \{\boldsymbol{r}_1, ..., \boldsymbol{r}_M\}$, where each $\boldsymbol{r}_j \in R$ is a row extracted from one of the time series $\boldsymbol{x}_i \in X$. Note that $M >> N$ and that the rows in $R$ are supposed to be i.i.d., i.e., we treat the samples in $R$ as independent from each other. Hence, the semantic space of our VAE ($\mathcal{Z} = \mathbb{R}^d$) will not embed any time-dependency among the rows of the same time series, since $\mathcal{E}_{\boldsymbol{\phi}}$ and $\mathcal{D}_{\boldsymbol{\varphi}}$ cannot observe this relation.

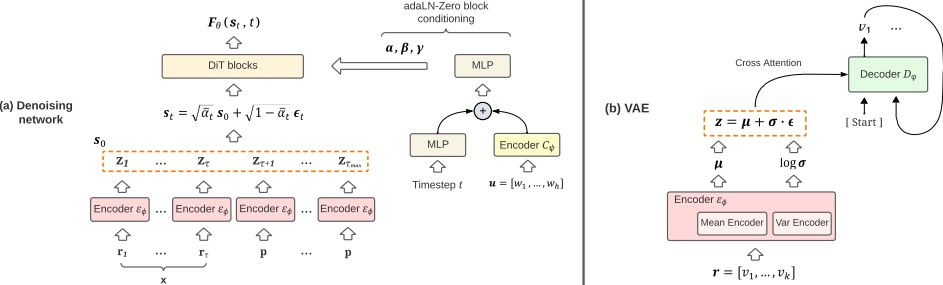

Figure 1: A schematic illustration of the denoising (a) and the VAE (b) network of TabDiT.

Fig. 1 shows the proposed framework, which includes $\mathcal{E}_{\boldsymbol{\phi}}$, $\mathcal{D}_{\boldsymbol{\varphi}}$ and our DiT-based denoising network $\mathcal{F}_{\boldsymbol{\theta}}$. At training time, the latter takes as input a sequence of latent row representations *of a specific time series*. In more detail, given $\boldsymbol{x}_i \in X$, $\boldsymbol{x}_i = [\boldsymbol{r}_1, ..., \boldsymbol{r}_\tau]$, we compute $\boldsymbol{s}_0 = [\boldsymbol{z}_1, ..., \boldsymbol{z}_\tau] = [\mathcal{E}_{\boldsymbol{\phi}}(\boldsymbol{r}_1), ..., \mathcal{E}_{\boldsymbol{\phi}}(\boldsymbol{r}_\tau)]$ and we use $\boldsymbol{s}_0$ as explained in Sec. 3 to train $\mathcal{F}_{\boldsymbol{\theta}}$, where $\mathcal{F}_{\boldsymbol{\theta}}(\boldsymbol{s}_t, t)$ implements $\epsilon_{\boldsymbol{\theta}}(\boldsymbol{s}_t, t)$. The timestep $t$ is first encoded using a small MLP and then its embedding vector is used to condition $\mathcal{F}_{\boldsymbol{\theta}}$ (see later). Note that $\boldsymbol{s}_0$ is a *time dependent* sequence of row embeddings, thus the real data distribution $q(\boldsymbol{s}_0)$ we use to train $\mathcal{F}_{\boldsymbol{\theta}}$ *does* include the statistical dependencies among tabular rows of a same time series. In other words, while $\mathcal{Z}$ represents only individual rows, we use $\mathcal{F}_{\boldsymbol{\theta}}$ to combine independent vectors lying in this space into a sequence of time-dependent final-embedding vectors representing an entire time series.

**Conditional generation.** In a conditional generation task, we want to condition $\mathcal{F}_{\boldsymbol{\theta}}$ using a row $\boldsymbol{u}$ of a parent table $P$ (Sec. 3). To do so, we first encode $\boldsymbol{u}$ using a specific encoder $\mathcal{C}_{\boldsymbol{\psi}}$. The architecture and the field value representations of $\mathcal{C}_{\boldsymbol{\psi}}$ are the same as $\mathcal{E}_{\boldsymbol{\phi}}$ (Sec. 4.1). However, $\mathcal{C}_{\boldsymbol{\psi}}$ is disjoint from the VAE and it is trained jointly with $\mathcal{F}_{\boldsymbol{\theta}}$. The output vector $\boldsymbol{c} = \mathcal{C}_{\boldsymbol{\psi}}(\boldsymbol{u})$ is summed with the embedding vector of the timestep $t$ and fed to the MLP of the adaLN-Zero block

conditioning mechanism (Sec. 3). Moreover, following most of the image generation DM literature, in the conditional generation scenario we also use Classifier-Free Guidance (CFG). Specifically, if the denoising network output is interpreted as the score function (Song et al., 2021), then the DM conditional sampling procedure can be formulated as (Peebles & Xie, 2023):

$$\hat{\mathcal{F}}_{\boldsymbol{\theta}}(\boldsymbol{s}_t, t, \boldsymbol{u}) = \mathcal{F}_{\boldsymbol{\theta}}(\boldsymbol{s}_t, t, \emptyset) + s \cdot (\mathcal{F}_{\boldsymbol{\theta}}(\boldsymbol{s}_t, t, \boldsymbol{u}) - \mathcal{F}_{\boldsymbol{\theta}}(\boldsymbol{s}_t, t, \emptyset)), \tag{2}$$

where $\mathcal{F}_{\boldsymbol{\theta}}(\boldsymbol{s}_t, t, \emptyset)$ is the unconditional prediction of the network and $s > 1$ indicates the scale of the guidance ($s = 1$ corresponds to no CFG). At training time, for each sample $\boldsymbol{x}_i$, with probability $p_d \in [0, 1]$ the condition $\boldsymbol{u}$ is dropped and the network is trained unconditionally.

### 4.1 Encoding and decoding in the VAE latent space

**Field value representations.** For categorical attributes $a_j$, we use a widely adopted tokenization approach (Zhang et al., 2024; Luetto et al., 2023; Padhi et al., 2021), in which each possible value $v_j \in V_j$ is associated with a token and a lookup table of token embeddings transforms these tokens into the initial embedding vectors of $\mathcal{E}_{\boldsymbol{\phi}}$. However, how to represent numerical values ($v_j \in \mathbb{R}$) in a way which is coherent with categorical feature tokens is still an open problem (Sec. 2) and each method adopts a specific solution (see App. A). For instance, REaLTabFormer (Solatorio & Dupriez, 2023) converts $v_j$ into a sequence of digits and then treats each digit as a categorical feature. If the maximum possible value in $X$ for the attribute $a_j$ is $v_{max_j}$, and assuming, for simplicity, that $a_j$ can only take on positive integer values, then $v_j$ is converted into a sequence $L = [D_1, ..., D_p]$, where each $D_k \in \{'0', ..., '9'\}$ corresponds to a digit in the decimal representation of $v_j$ and $p$ is a fixed sequence length corresponding to the number of digits necessary to represent $v_{max_j}$. Importantly, if the decimal representation of $v_j$ requires less than $p$ digits, the sequence is left-padded with zeros (Solatorio & Dupriez, 2023). This representation is lossless (App. A), but it leads to very long sequences which are frequently full of zeros. Indeed, the value distribution for $a_j$ is usually a Gaussian with very long tails. For instance, the "amount" field of a bank transaction can range from tens of millions to a few cents, but most values are smaller than 1,000 (e.g., 35$).

To solve this problem, we propose a *variable-range* representation using a small, fixed number of digits preceded by a magnitude order. For simplicity, let us assume that $v_j$ is a positive integer, thus:

$$v_j = \sum_{k=0}^{m} b_k 10^k, \quad DR(v_j) = [b_m b_{m-1}...b_0], \tag{3}$$

where $m$ is the largest exponent and $DR(v_j)$ is the representation of $v_j$ by means of a sequence of digits (e.g., [35967]). Then, we represent $v_j$ using a sequence $Q$ defined as follows:

$$Q = [O, D_m, D_{m-1}, ..., D_{m-n+1}]. \tag{4}$$

In Eq. (4), $O$ is the *magnitude order prefix* and it corresponds to $m$ in Eq. (3). Specifically, in the tabular data domain we can assume that $v_j < 10^{10}$, hence, $m \in \{0, ..., 9\}$ and $O \in \{'0', ..., '9'\}$ is the token corresponding to $m$. The value of $O$ is the first one that will be generated by $\mathcal{D}_{\boldsymbol{\varphi}}$ when decoding the sequence representing a numerical value, which corresponds to predict its magnitude order ($m$). We then encode the $n$ most significant digits of $v_j$ using $D_m, D_{m-1}, ..., D_{m-n+1}$, where $D_k \in \{'0', ..., '9'\}$ ($m \le k \le m - n + 1$) is the token corresponding to the digit $b_k$, and $[b_m b_{m-1}...b_{m-n+1}] \subseteq DR(v_j)$ is the ($m$-depending) range we represent. For instance, if $v_j = 35967$ and $n = 4$, then $Q = ['4', '3', '5', '9', '6']$. Once decoded, $Q$ can be used to compute the (possibly truncated) value of $v_j$. For instance, using $Q = ['4', '3', '5', '9', '6']$, we get: $\hat{v}_j = 3 * 10^4 + 5 * 10^3 + 9 * 10^2 + 6 * 10 = 35960$. We use $n = 4$ *regardless of attribute or dataset*, and this value was chosen in preliminary studies as a trade-off between the length of the resulting sequences $Q$ and the amount of truncated information (more details in App. A). On the other hand, if $m < n$ (the most frequent case), no truncation is necessary and we right-pad $Q$ with zeros. For instance, if $v_j = 35$, then we use $Q = ['1', '3', '5', '0', '0']$. At decoding time, $O$ is the first token generated by $\mathcal{D}_{\boldsymbol{\varphi}}$, which is converted into $m$. If $m < n$, the last $n - m - 1$ tokens in $Q$ are ignored because they are zero-padding digits (e.g., in $Q = ['1', '3', '5', '0', '0']$, this corresponds to

the last 2 elements of $Q$). This implies that possible generation errors in the last $n - m - 1$ tokens are ignored. More formally, using our representation, the joint distribution over the tokens that the decoder should model is restricted to $min(n + 1, m + 2)$ variables, as opposed to a joint distribution over $p$ variables ($p > m + 2$) as in the case of the fixed digit sequence proposed in (Solatorio & Dupriez, 2023), reducing the overall error probability (see App. A for more details).

**Encoder and Decoder Architectures.** Given a tabular row $r$, each categorical value is converted into a token and each numerical value is converted in a sequence $Q$ of $n + 1$ tokens (see above). After that, we use attribute-specific lookup tables to represent all the tokens as embedding vectors: $e_1, ... e_\nu$, where $\nu$ is the sum of the number of categorical attributes plus the number of the numerical attributes multiplied by $(n + 1)$. The whole sequence $e_1, ... e_\nu$ is fed to $\mathcal{E}_\phi$, which, following (Zhang et al., 2024), is composed of two separated towers, respectively computing the mean ($\mu$) and log variance ($\log \sigma$) of the latent representation $z$ of $r$ in $\mathcal{Z}$. However, differently from (Zhang et al., 2024), we use multi-head self-attention in $\mathcal{E}_\phi$ because, in preliminary experiments, we found that this is more effective than single-head attention.

Using the standard VAE reparameterization trick, we obtain: $z = \mu + \sigma \cdot \epsilon, \epsilon \sim \mathcal{N}(0, I)$, and $z$ is fed to the decoder $\mathcal{D}_\varphi$. Moreover, differently from most DM-based tabular data generation approaches, we propose an AR decoder which is conditioned on $z$. Specifically, decoding a tabular row $r$ starts with a special token [Start]. Every time the next-token is predicted, it is coded back and fed to $\mathcal{D}_\varphi$, which is a Transformer with 3 blocks. Each block alternates multi-head causal attention layers with respect to previously predicted tokens with (multi-head) cross attention layers to the $\nu$ embedding vectors of $z$. For each categorical attribute $a_j$, a final linear layer followed by softmax computes a posterior over the specific vocabulary $V_j$. Similarly, if $a_j$ is numerical, it computes a posterior for each of the $n + 1$ tokens in its variable-range representation $Q$. At inference time, we *deterministically* select a token from each of these posteriors using arg max. Finally, the predicted value $v_j$ is autoregressively encoded into $\mathcal{D}_\varphi$ using the same field value representation mechanism used for the encoder (more details in App. F).

### 4.2 VARIABLE-LENGTH TIME SERIES

Given two sequences $x_i = [r_1, ..., r_{\tau_i}]$ and $x_j = [r_1, ..., r_{\tau_j}]$, $x_i, x_j \in X$, their length is usually different ($\tau_i \neq \tau_j$). Following FiT (Lu et al., 2024), we use a maximum length $\tau_{max}$ and, for each $x_i$, we append $\tau_{max} - \tau_i$ *padding rows* $p$ (see below) to the right side of $x_i$. However, in FiT the padding tokens are used only to pack data into batches of uniform shape for parallel processing, and are ignored during the forward pass using a masked attention (basically, they are not used). Conversely, we use our padding rows to let the network decide the length of the time series it is generating. To do so, we add an [EoS] token to each categorical vocabulary $V_j$, included the vocabulary representing the digits, and we form the special row $p = [[EoS], ..., [EoS]]$. During VAE training, with probability 0.05 we sample $p$ and with 0.95 we sample a real row from $R$. During the denoising network training, if $\tau_i < \tau_{max}$, $x_i$ is padded with $p$, which results in $s_0$ being a sequence of $\tau_{max}$ latent vectors, where the last vectors correspond to the representation of $p$ in $Z$ and contribute to compute the loss. Finally, at inference time, we randomly sample $\tau_{max}$ vectors $z_1, ..., z_{\tau_{max}}$ in $\mathcal{Z}$, which form $s_T = [z_1, ..., z_{\tau_{max}}]$, the starting point of the DM sampling chain. The ending point of the sampling chain $s_0$ is split in $\tau_{max}$ final vectors $z'_1, ..., z'_{\tau_{max}}$ which are individually decoded using $\mathcal{D}_\varphi$. We stop decoding as soon as we meet the first padding row $p$. More precisely, we stop decoding when we meet the first row containing at least one token [EoS]. In this way, we use $\mathcal{F}_\theta$ to *predict the end of the time series jointly with its content*.

## 5 EXPERIMENTS

### 5.1 EVALUATION PROTOCOL

Due to the lack of a shared evaluation protocol for tabular data time series generation, we propose a unified framework which is composed of different public datasets, conditional and unconditional generation tasks and different metrics. More details are provided in App. C and E.

**Datasets.** We use six public datasets, whose statistics are provided in App. E. *Age1*, *Age2*, *Leaving*, taken from (Fursov et al., 2021), and *PKDD'99 Financial Dataset*, taken from (Berka, 1999), are composed of bank transaction time series of different real banks with different attributes. Each time

series is the temporally ordered sequence of bank transactions of a given bank client. On the other hand, the *Rossmann* and the *Airbnb* datasets, used in (Patki et al., 2016; Solatorio & Dupriez, 2023), are composed of, respectively, historic sales data for different stores and access log data from Airbnb users. These six datasets are widely different from each other both in terms of their attributes and their sizes, thus representing very different application scenarios. Two of these datasets, *Age1* and *Leaving*, do not have an associated parent table, so they are used only for unconditional generation.

**Metrics.** Since heterogeneous time series generation is a relatively unexplored domain, there is also a lack of consolidated metrics. For instance, Solatorio & Dupriez (2023) use the *Logistic Detection*, which is a discriminative metric based on training a binary classifier to distinguish between real and generated data, and then using the ROC-AUC scores of the discriminator (the higher the better ↑) on an held-out set of real and synthetic data (App. C). We also adopt this metric in Sec. 5.3 to compare our results with REaLTabFormer. However, the classifier used in the Logistic Detection (a Random Forest) takes as input only an *individual row* of the real/generated time series, thus this criterion is insufficient to assess, e.g., the temporal coherence of a time series composed of different rows (e.g., see App. H.3 for a few examples of time series generated by REaLTabFormer with an incoherent sequence of dates). For this reason, in this paper we extend this metric using a classifier which takes as input the *entire time series* instead of individual rows. Inspired by similar metrics commonly adopted in single-row tabular data generation (Liu et al., 2023), we call our metric Machine Learning Detection (*MLD*) and we measure the discriminator accuracy (the lower the better ↓, because it means that the discriminator struggles in separating the real from the synthetic distribution). To emphasize the difference with respect to the Logistic Detection used in Solatorio & Dupriez (2023), use the suffix "SR" (single row) for the latter (*LD-SR*) and the suffix "TS" (*MLD-TS*) to indicate that our MLD metric depends on the entire time series. More specifically, in *MLD-TS* we use CatBoost (Prokhorenkova et al., 2018) as the classifier, because it is one of the most common non-deep learning based methods for heterogeneous tabular data discriminative tasks (Luetto et al., 2023; Gorishniy et al., 2021; Chen & Guestrin, 2016), jointly with a standard library (Christ et al., 2018) extracting a fixed-size feature vector from a time series (Luetto et al., 2023). We provide more details on these metrics in App. C, where we also introduce an additional metric based on the Machine Learning Efficiency (MLE) (Zhang et al., 2024; Kotelnikov et al., 2023).

## 5.2 ABLATION

In Tab. 1 we use *Age2*, which, among the six datasets, is both one of the largest in terms of number of time series and one of the most complex for different types of attributes. The results in Tab. 1 are based on an unconditional generation task and evaluate the contribution of each component of our method. The last row, *TabDiT*, refers to the full method as described in Sec. 4, while all the other rows refer to our full-model with one missing component. For instance, *Parallel VAE* refers to a standard, non-AR VAE decoder, in which all the fields of a tabular row are predicted in parallel. *No cross-att VAE* differs from the VAE introduced in Sec. 4.1 because in $\mathcal{D}_\varphi$ we remove the cross attention layers to $z$, and $z$ is directly fed to $\mathcal{D}_\varphi$ in place of the [Start] token. In all the other entries of the table we use our AR VAE as described in Sec. 4.1. *Fixed digit seq* corresponds to the numerical value representation proposed in REaLTabFormer (Solatorio & Dupriez, 2023), in which we use $p = 7$ digits (Sec. 4.1), which are enough to represent all the numerical values in *Age2*. In *Quantize*, we follow TabGPT (Padhi et al., 2021) and we convert each numerical feature into a categorical one, using quantization and a field specific vocabulary. In *Linear transf*, we follow Tab-Syn (Zhang et al., 2024), where numerical features are predicted using a linear transformation of the corresponding last-layer token embedding of the VAE decoder. We refer to App. A for more details on these representations. In all the variants, we always use a coherent numerical value representation in the corresponding VAE encoder. We also evaluated hybrid solutions, where VAE encoders and decoders have different representations of the numerical values, but we always obtained worse results than in cases of coherent representations. Moreover, in *W/o length pred* we follow FiT (Lu et al., 2024) and we use a masked attention which completely ignores the padding rows. In this case, at testing time the sequence length is randomly sampled using a mono-modal Gaussian distribution fitted on the length of the training time series. Finally, *AR Baseline* is a (strong) baseline based on a purely AR Transformer which we use to validate the effectiveness of the DM paradigm (see below).

In our **AR Baseline**, we modify the *hierarchical discriminative* architecture proposed in (Padhi et al., 2021) and adopted also in (Luetto et al., 2023) to create an AR generative model. Specifically, we use a causal attention in the "Sequence Transformer" and we replace the "Field Transformer"

Table 1: Ablation study on the *Age2* dataset.

| Method | Parallel VAE | No cross-att VAE | Fixed digit seq | Quantize | Linear transf | W/o length pred | AR Baseline | TabDiT |
|---|---|---|---|---|---|---|---|---|
| **MLD-TS** ↓ | 62.2 | 51.2 | 64.0 | 83.8 | 83.7 | 61.0 | 68.3 | **50.8** |

with one of the towers of our VAE encoder architecture. Moreover, we use our AR VAE decoder architecture to predict the output sequence. Note that we use the VAE architectural components but we *do not* use variational training and the entire network is trained end-to-end using only a next-token prediction task, following (Padhi et al., 2021; Solatorio & Dupriez, 2023). The numerical features are represented using our variable-range decimal representation. In short, this baseline is a purely AR Transformer where we merged the hierarchical architecture used in (Padhi et al., 2021; Luetto et al., 2023) with our decoding scheme and our feature value representation.

Tab. 1 shows that all the components of the proposed method are important, since their individual removal always leads to a significant decrement of the MLD. Specifically, the numerical value representation has a high impact on the results. For instance, both *Linear transf*, adopted in (Zhang et al., 2024), and *Quantize*, used in (Padhi et al., 2021), lead to a drastic worsening of results. On the other hand, our *AR Baseline*, which is based on the same input representation and shares important architectural details with TabDiT, is largely outperformed by the latter, showing the advantage of our LDM-based approach. Finally, the difference between *No cross-att VAE* and the full method is subtle, showing that the cross attention layers in $\mathcal{D}_\varphi$ can be replaced by directly feeding $z$ to the decoder, as long as the latter has an AR architecture (see App. D.3 for additional ablation experiments).

### 5.3 MAIN EXPERIMENTS

Following the protocol adopted in (Solatorio & Dupriez, 2023), all the experiments of this section have been repeated three times with different random splits of the samples between the generator training data, the discriminators' training data and the testing data (see App. C and E for more details). For each experiment, we report the means and the standard deviations of the *MLD-TS* and the *LD-SR* metrics. In App. D.1 we show additional experiments using other metrics App. C.

**Unconditional generation.** We are not aware of any unconditional generative model for time series of heterogeneous tabular data with public code or published results. Indeed, REaLTabFormer is a *conditional* method, while TabGPT is a *forecasting* model, both lacking of an unconditional sampling mechanism (Sec. 1 and 2). For this reason, we can only compare TabDiT with our AR Baseline (Sec. 5.2). The unconditional results in Tab. 2 and 3 show that TabDiT outperforms the AR Baseline in all the datasets by a *large margin*. For instance, in the largest dataset (*Age1*), TabDiT improves the *MLD-TS* score by more than 27 points compared to the AR baseline. On *Age2*, TabDiT outperforms the AR Baseline by more than 17 points, achieving an almost ideal situation in which the real and the generated distributions cannot be distinguished by each other (discrimination accuracy = 50.43%, very close to the chance level). Note that the *Age2* results are slightly different from those reported in Tab. 1 because they were obtained averaging 3 different runs.

**Conditional generation.** We indicate with "child gt-cond" the conditional generation task in which the parent table row $u$, used for conditioning, is a ground truth, real row extracted from the testing dataset (Sec. 3). Specifically, we use all the elements $u \in P_{test}$ (Sec. 3) to condition the generator networks. Moreover, to make a comparison with Solatorio & Dupriez (2023) possible, we also follow their protocol and we indicate with "child" the conditional generation task on the time series ($x$) where also the conditioning information ($u$) is automatically generated. Specifically, in our case, we generate $u$ by training a dedicated DiT-based denoising network and a corresponding AR VAE on $P$. These networks have the same structure and are trained using the same approach described in Sec. 4 but using $P$ instead of $X$ (more details in App. B). Finally, in (Solatorio & Dupriez, 2023) "merged" indicates the evaluation of the joint probability of generating both $x$ and $u$. Using *MLD-TS* and *LD-SR*, this is obtained by concatenating $u$ with either $x$ or with its individual rows $r$, respectively, and then feeding the result to the corresponding discriminator (see App. C). In Tab. 4, we indicate with * the results of REaLTabFormer and SDV which we report from (Solatorio & Dupriez, 2023), while in all the other cases they have been reproduced by us using the corresponding public available code. Specifically, while using REaLTabFormer with a real data-conditioning task ("child gt-cond")

is easy, that was not possible for SDV. The choice of REaLTabFormer and SDV follows (Solatorio & Dupriez, 2023), where the selected baselines are those which have open-sourced models.

The results in Tab. 4 confirm the results in Tab. 2 and 3. Across all datasets and tasks, and with both metrics, TabDiT outperforms all the baselines by a large margin, often approaching the lower bound of $50\%$ *MLD-TS* accuracy. In the "child gt-cond" task, AR Baseline is the second best most of the time, while in "child" and "merged" the second best is REaLTabFormer. We believe that the reason of this discrepancy is most likely due to the fact that both the "child" and the "merged" task evaluation depend on the quality of the *single-row* parent generation, in which REaLTabFormer gets better results (see App. B). Finally, in App. D.1 we present additional experiments using the MLE (Sec. 5.1), where we also show how the generated data can be effectively used to replace real data for classification tasks (Sec. 1), and in App. D.2 we extend the results of this section to larger datasets.

Table 2: Unconditional generation results on *Rossmann*, *Airbnb* and *PKDD'99*.

| | Rossmann | | Airbnb | | PKDD'99 Financial | |
|---|---|---|---|---|---|---|
| Method | MLD-TS ↓ | LD-SR ↑ | MLD-TS ↓ | LD-SR ↑ | MLD-TS ↓ | LD-SR ↑ |
| AR Baseline (ours) | $97.80_{\pm2.20}$ | $49.97_{\pm3.26}$ | $77.23_{\pm1.46}$ | $56.43_{\pm2.80}$ | $92.87_{\pm1.68}$ | $71.93_{\pm1.34}$ |
| TabDiT (ours) | $\mathbf{82.60}_{\pm3.92}$ | $\mathbf{77.07}_{\pm5.37}$ | $\mathbf{55.07}_{\pm3.52}$ | $\mathbf{78.07}_{\pm2.77}$ | $\mathbf{85.53}_{\pm4.18}$ | $\mathbf{79.10}_{\pm6.09}$ |

Table 3: Unconditional generation results on *Age2*, *Age1* and *Leaving*.

| | Age2 | | Age1 | | Leaving | |
|---|---|---|---|---|---|---|
| Method | MLD-TS ↓ | LD-SR ↑ | MLD-TS ↓ | LD-SR ↑ | MLD-TS ↓ | LD-SR ↑ |
| AR Baseline (ours) | $67.53_{\pm0.75}$ | $83.47_{\pm0.38}$ | $91.20_{\pm0.46}$ | $74.23_{\pm1.06}$ | $69.43_{\pm4.02}$ | $75.33_{\pm2.86}$ |
| TabDiT (ours) | $\mathbf{50.43}_{\pm1.85}$ | $\mathbf{87.00}_{\pm1.54}$ | $\mathbf{63.93}_{\pm3.20}$ | $\mathbf{76.00}_{\pm4.25}$ | $\mathbf{62.33}_{\pm0.99}$ | $\mathbf{75.63}_{\pm4.20}$ |

Table 4: Conditional generation results. * Values reported from Solatorio & Dupriez (2023).

| Method | Task | Rossmann | | Airbnb | | Age2 | | PKDD'99 Financial | |
|---|---|---|---|---|---|---|---|---|---|
| | | MLD-TS ↓ | LD-SR ↑ | MLD-TS ↓ | LD-SR ↑ | MLD-TS ↓ | LD-SR ↑ | MLD-TS ↓ | LD-SR ↑ |
| SDV | child | $99.63_{\pm0.64}$ | $6.53^*_{\pm0.39}$ | $93.30_{\pm0.61}$ | $0.00^*_{\pm0.00}$ | $96.03_{\pm0.11}$ | $44.80_{\pm1.73}$ | $97.95_{\pm1.42}$ | $6.53_{\pm0.58}$ |
| | merged | $100.00_{\pm0.00}$ | $2.80^*_{\pm0.25}$ | $94.40_{\pm1.65}$ | $0.00^*_{\pm0.00}$ | $96.27_{\pm0.06}$ | $37.63_{\pm1.47}$ | $98.12_{\pm1.17}$ | $8.77_{\pm0.59}$ |
| REaLTabFormer | child gt-cond | $98.90_{\pm1.10}$ | $60.63_{\pm2.65}$ | $63.63_{\pm1.20}$ | $\underline{86.17}_{\pm1.29}$ | $\underline{66.77}_{\pm0.42}$ | $77.90_{\pm0.85}$ | $97.87_{\pm0.59}$ | $21.97_{\pm0.55}$ |
| | child | $\underline{64.83}_{\pm1.33}$ | $\underline{52.08}\,^*_{\pm0.89}$ | $\underline{57.77}_{\pm0.67}$ | $30.48^*_{\pm0.79}$ | $\underline{52.97}_{\pm2.32}$ | $\underline{77.30}_{\pm0.92}$ | $\underline{59.33}_{\pm3.82}$ | $21.50_{\pm0.72}$ |
| | merged | $\underline{74.43}_{\pm8.85}$ | $\underline{28.33}\,^*_{\pm2.31}$ | $\underline{76.97}_{\pm2.04}$ | $\underline{21.43}\,^*_{\pm1.10}$ | $\underline{52.10}_{\pm2.17}$ | $\underline{75.53}_{\pm0.65}$ | $\underline{58.77}_{\pm3.05}$ | $26.00_{\pm1.61}$ |
| AR Baseline (ours) | child gt-cond | $\underline{95.57}_{\pm1.96}$ | $\underline{71.60}_{\pm2.42}$ | $\underline{57.97}_{\pm1.72}$ | $82.77_{\pm0.49}$ | $69.97_{\pm0.90}$ | $\underline{80.73}_{\pm0.59}$ | $\underline{68.33}_{\pm4.37}$ | $\underline{81.13}_{\pm1.51}$ |
| | child | $99.63_{\pm0.64}$ | $36.03_{\pm8.79}$ | $82.33_{\pm1.53}$ | $\underline{62.53}_{\pm3.93}$ | $79.03_{\pm1.62}$ | $65.83_{\pm3.95}$ | $79.07_{\pm6.23}$ | $\underline{67.60}_{\pm4.36}$ |
| | merged | $99.63_{\pm0.64}$ | $19.70_{\pm6.80}$ | $93.50_{\pm1.30}$ | $8.53_{\pm2.49}$ | $81.30_{\pm0.78}$ | $48.03_{\pm3.61}$ | $83.33_{\pm5.75}$ | $\underline{38.73}_{\pm4.22}$ |
| TabDiT (ours) | child gt-cond | $\mathbf{72.20}_{\pm1.10}$ | $\mathbf{82.90}_{\pm1.32}$ | $\mathbf{51.10}_{\pm2.60}$ | $\mathbf{98.07}_{\pm0.25}$ | $\mathbf{51.40}_{\pm2.95}$ | $\mathbf{84.60}_{\pm1.87}$ | $\mathbf{59.50}_{\pm10.53}$ | $\mathbf{81.20}_{\pm2.71}$ |
| | child | $\mathbf{64.03}_{\pm0.64}$ | $\mathbf{80.13}_{\pm3.02}$ | $\mathbf{49.33}_{\pm1.18}$ | $\mathbf{81.10}_{\pm0.98}$ | $\mathbf{50.47}_{\pm1.71}$ | $\mathbf{84.70}_{\pm1.21}$ | $\mathbf{51.80}_{\pm6.44}$ | $\mathbf{79.13}_{\pm3.04}$ |
| | merged | $\mathbf{71.83}_{\pm2.77}$ | $\mathbf{38.63}_{\pm1.04}$ | $\mathbf{54.63}_{\pm0.85}$ | $\mathbf{47.37}_{\pm2.68}$ | $\mathbf{51.53}_{\pm3.04}$ | $\mathbf{78.93}_{\pm0.64}$ | $\mathbf{54.03}_{\pm3.95}$ | $\mathbf{53.20}_{\pm0.66}$ |

## 6 CONCLUSIONS

We presented TabDiT, an LDM approach for tabular data time series generation. Differently from the common LDM paradigm, where an entity domain is holistically represented in the VAE latent space, we split our time series in individual tabular rows, which are compressed independently one from the others. This simplifies the variational learning task and avoids the need to represent variable length sequences in the VAE latent space. Then, a DiT denoising network combines embedding vectors defined in this space into final sequences, in this way modeling their temporal dynamics. Moreover, our DiT explicitly predicts the end of the generated time series to jointly model its content and its length. Furthermore, we proposed an Autoregressive VAE decoder and a variable-range decimal representation of the numerical values to encode and decode heterogeneous field values. Using extensive experiments with six different datasets, we showed the effectiveness of TabDiT, which largely outperfoms the other baselines in both conditional and unconditional tasks. Specifically, TabDiT is the first network showing results for unconditional generation of time series of tabular data with heterogeneous field values.

ACKNOWLEDGMENTS

This work was partially supported by the EU Horizon project ELIAS (No. 101120237). We thank the Data Science team of Prometeia S.p.A. for their valuable collaboration and for providing useful discussions and insights that contributed to this work.

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

## A  NUMERICAL VALUE REPRESENTATIONS

In this section, we provide more details on the numerical value representation approaches used by previous work and in our ablation analysis in Sec. 5.2, as well as on our variable-range representation. For clarity, we adopt the same terminology and numerical examples used in Sec. 4.1.

TabGPT (Padhi et al., 2021) applies a quantization which associates $v_j$ with a bin value $B$, which is then treated as a categorical feature. The disadvantage of this representation is an information loss due to the difference between the decoded bin $B$ and the actual value $v_j$. Indeed, $B$ corresponds to the center of the numerical interval assigned to the $B$-th bin during the quantization phase. This coding-decoding scheme corresponds to the entry *Quantize* in Tab. 1.

TabSyn (Zhang et al., 2024) applies a linear transformation when coding $v_j$ and another linear transformation (i.e., a linear regression) when decoding back the embedding vector of the last-layer decoder to $v_j$. However, also this solution is sub-optimal, because linear regression struggles in respecting some implicit value distribution constraints. For instance, if the admissible values for $a_j$ are only integers, a linear regression layer may predict a number with a decimal point. This coding-decoding scheme corresponds to the entry *Linear transf* in Tab. 1.

Finally, *Fixed digit seq* in Tab. 1 corresponds to the sequence of digits $L$ (see Sec. 4.1) used in REaLTabFormer (Solatorio & Dupriez, 2023). We believe that one of the reasons why this representation is sub-optimal with respect to our variable-range decimal representation $Q$ (Sec. 4.1 and 5.2) is that its longer length leads, on average, to a more difficult decoding problem. For instance, in the *Age2* dataset, the "amount" attribute needs $p = 7$ digits to represent $v_{max_j}$. Thus, the length of $L$ is $p = 7$. When the VAE decoder should decode a numerical value, the probability of error is given by the (complement of the) joint distribution of all the digits of its representation $L$. For instance, if the target value is $v_j = 35$, then, the decoder should generate this sequence: $L = ['0', '0', '0', '0', '0', '3', '5']$. The probability of error when generating $L$ is:

$$\mathcal{P}_L = 1 - P_L(L) = 1 - \prod_{k=1}^{p} P_L(D_k | D_1, ..., D_{k-1}), \tag{5}$$

which, in case of $v_j = 35$, is:

$$1 - [P_L('0')P_L('0'|'0')P_L('0'|'0', '0')P_L('0'|'0', '0', '0')$$
$$P_L('0'|'0', '0', '0', '0')P_L('3'|'0', '0', '0', '0', '0')$$
$$P_L('5'|'3', '0', '0', '0', '0', '0')]. \tag{6}$$

On the other hand, in case of $Q$ (Sec. 4.1), since we use an AR decoding, once the magnitude order prefix $O$ has been generated, we can convert $O$ in its corresponding value $m$ (Eq. (3)) and use $m$ to *ignore* possible zero-padding tokens on the right side of the sequence. Specifically, if $m < n$, the probability of error is given by:

$$\mathcal{P}_Q = 1 - P_Q(Q) = 1 - [P_Q(O) \prod_{k=0}^{m} P_Q(D_{m-k}|O, D_m, ..., D_{m-k+1})], \tag{7}$$

which, in case of $v_j = 35$, $Q = ['1', '3', '5', '0', '0']$, and $m = 1$, is:

$$1 - [P_Q('1')P_Q('3'|'1')P_Q('5'|'1', '3')]. \tag{8}$$

The comparison between Eq. (6) and Eq. (8) intuitively shows that $Q$ is much easier to generate than $L$ if $v_j$ is small. More formally, if we assume that all the digit generations have, on average, the same probability to be correct, i.e., that, on average: $P_Q(O) = P_L(D_1)$ and $P_L(D_k|D_1, ..., D_{k-1}) = P_Q(D_{m-k}|O, D_m, ..., D_{m-k+1})$, then, if $m < p - 2$, from Eq. (5) and (7) follows that $\mathcal{P}_L > \mathcal{P}_Q$.

The proposed representation can be easily extended to negative numbers and non-integer values. For instance, if the admissible values for $a_j$ include negative numbers, then we prepend in $Q$ a token $S$ corresponding to the sign of $v_j$: $Q = [S, O, D_m, D_{m-1}, ..., D_{m-n+1}]$, where $S \in \{'-', '+'\}$. Note that, in case of negative numbers, also $L$ should be extended to include $S$ (Solatorio & Dupriez, 2023). On the other hand, in case of rational numbers, $L$ needs to be extended to include a token representing the decimal point (Solatorio & Dupriez, 2023), while our variable-range representation $Q$ remains unchanged. For instance, if $v_j = 3.5$, then we have $Q = ['0', '3', '5', '0', '0']$.

Finally, we provide below more details on how $n$ was chosen. We used the *Age2* dataset (Fursov et al., 2021), adopted for most of our ablation studies. For each numerical values $v_j$ in *Age2*, we compute $DR(v_j)$ (Eq. (3)) and we remove from $DR(v_j)$ the possible subsequence of only zeros on its right (e.g., from [87600] we remove [00]). Note that these all-zero subsequences do not lead to any truncation error. Let *Significant*$(v_j)$ be the part of $DR(v_j)$ that remains after this cut (e.g., in the previous example, *Significant*$(v_j) = [876]$). Finally, we computed the average ($\mu_S$) and the standard deviation ($\sigma_S$) of the lengths of all the sequences *Significant*$(v_j)$ in the training dataset, getting $\mu_S = 2.26$ and $\sigma_S = 0.47$, respectively. The value $n = 4$ was chosen as the first integer greater than $\mu_S + 2\sigma_S$. In this way, most of the training data in *Age2* do not need any truncation when represented using $Q$. The value $n = 4$ was used in all the other datasets. In App. H.1 we show some qualitative results which compare to each other the distributions of some numerical values generated using the representations presented in this section.

## B  TABDIT FOR SINGLE TABULAR ROW GENERATION

Generating single tabular rows is out of the scope of this paper, in which we focus on the (more challenging) generation of time series. However, following the protocol proposed in (Solatorio & Dupriez, 2023), in Sec. 5.3 we show conditional experiments where the "parent" row table $\boldsymbol{u}$ is automatically generated (tasks "child" and "merged"). In order to generate a row $\boldsymbol{u}$ following the empirical distribution in $P$ (Sec. 3), we adapted our TabDiT to a single row generation task. Specifically, we train a parent-table *dedicated* VAE encoder $\mathcal{E}_{\boldsymbol{\phi}_P}^P$ and decoder $\mathcal{D}_{\boldsymbol{\varphi}_P}^P$, as well as a *dedicated* DiT-based denoising network $\mathcal{F}_{\boldsymbol{\theta}_P}^P$. These networks all have the same structure as those used for time series generation (Sec. 4), including the same number of layers and parameters. The only difference is that the sequence of embedding vectors fed to $\mathcal{F}_{\boldsymbol{\theta}_P}^P$ corresponds to *individual field*

Table 5: Parent generation results using the **LD-SR** ↑ metric (mean and standard deviation over three runs). * These values are reported from Solatorio & Dupriez (2023).

| Method | Rossmann | Airbnb | Age2 | PKDD'99 Financial |
|---|---|---|---|---|
| SDV | 31.77* ±3.41 | 7.37* ±0.72 | 55.97 ±2.14 | 37.87 ±2.59 |
| REaLTabFormer | 81.04 * ±4.54 | **89.65*** ±1.92 | **98.13** ±2.73 | **98.70** ±2.17 |
| AR Baseline (ours) | 52.47 ±9.31 | 13.33 ±3.12 | 63.03 ±3.36 | 62.97 ±8.13 |
| SR-TabDiT (ours) | **91.60** ±3.80 | 84.53 ±0.45 | 91.90 ±0.92 | 81.77 ±4.72 |

*value embeddings* of a single tabular row in the VAE latent space. Specifically, if $\boldsymbol{y} = \mathcal{E}_{\boldsymbol{\phi}_P}^P(\boldsymbol{u})$ is the latent representation of $\boldsymbol{u} = [w_1, ..., w_h] \in P$, we split $\boldsymbol{y}$ in $h$ separate vectors, $\boldsymbol{y}_1, ..., \boldsymbol{y}_h$, corresponding to the final embeddings of the *field values* $w_1, ..., w_h$ in the last-layer of $\mathcal{E}_{\boldsymbol{\phi}_P}^P$. Then, the sequence of embeddings used in the forward process for training $\mathcal{F}_{\boldsymbol{\theta}_P}^P$ is $\boldsymbol{s}_0 = [\boldsymbol{y}_1, ..., \boldsymbol{y}_h]$. The rest of the training and sampling procedures follows the method presented in Sec. 4. We call this method Single Row TabDiT (*SR-TabDiT*).

For the AR Baseline, $\boldsymbol{u}$ is simply concatenated on the left-side of $\boldsymbol{x}$, and we autoregressively generate the sequence $[\boldsymbol{u}, \boldsymbol{x}]$ starting from a [SoS] token. Tab. 5 shows the results on the "parent" (alone) generation task following the protocol used in (Solatorio & Dupriez, 2023). In most datasets, REaLTabFormer beats SR-TabDiT, which is the second best. However, as mentioned above, the goal of this paper is to propose a time series generation approach, and we have developed a single-row generation model only to make a comparison with REaLTabFormer possible using the "child" and the "merged" tasks proposed in that paper. Note also that we have *not* optimized our method for this task and, most likely, simply reducing the number of layers and parameters of SR-TabDiT may help in regularizing training. However, we leave the study of how to better adapt TabDiT to a single-row generation task as future work. Moreover, note that in Tab. 4, TabDiT largely outperforms REaLTab-Former also in the "child" and the "merged" tasks *despite it is conditioned on generated parent rows with a lower quality compared to REaLTabFormer*. Indeed, success on these tasks necessarily depends on the quality of the row $\boldsymbol{u}$ that was generated before the conditional process. Finally, we note that in Tab. 4, in most datasets the AR Baseline beats REaLTabFormer in the "child gt-cond" task, *in which the parent row is not generated*, but underperfoms REaLTabFormer in the "child" and the "merged" tasks. Most likely, the reason is that also our AR Baseline is disadvantaged due to a lower quality parent row generation.

## C  METRICS

Besides the discriminative metrics described in Sec. 5.1, in the single-row tabular data generation literature there are many other evaluation metrics, which however we do not believe suitable for the time series domain. For instance, *low-order statistics* include statistics computed either using the values of an individual tabular attribute or statistics such as the pair-wise correlation between the values of two numerical attributes (Zhang et al., 2024). However, in a time series composed of dozens of rows, each row composed of different fields, statistics on a single field or pairs of fields are not very informative. Similarly, we do not use *high-order statistics* (e.g., $\alpha$-precision and $\beta$-recall) (Zhang et al., 2024), because they are single-row criteria and they usually use a fragile nearest-neighbor like approach in the data space to estimate the distribution coverage.

On the other hand, we believe that the most useful metric is the *MLD-TS* proposed in Sec. 5.1, for which we provide below additional details. The CatBoost (Prokhorenkova et al., 2018) discriminator is trained using a balanced dataset composed of both real and synthetic data (separately generated by each compared generative method). Then, a *separate* testing set, also composed of 50% real and 50% generated data, is used to assess the discriminator accuracy. Real training and testing data *do not* include data used to train the generator. Thus, basically the real data are split in: samples used to train the generator, samples used (jointly with synthetic data) to train the discriminator, and samples used (jointly with other synthetic data) to test the discriminator. We randomly change these three splits in each of the run used to compute the results in Sec. 5.3.[1]

---

[1]The evaluation protocol code is available at: `https://github.com/fabriziogaruti/TabDiT`

We use the same training-testing protocol for the *Logistic Detection*, which, following (Solatorio & Dupriez, 2023), is defined as: $LD\text{-}SR = 100 \times (1 - \mu_{RA})$, where:

$$\mu_{RA} = \frac{1}{F} \sum_{i=1}^{F} \max(0.5, ROC - AUC) \times 2 - 1. \tag{9}$$

In Eq. (9), ROC and AUC indicate the ROC-AUC scores, computed using a Random Forest trained and tested using single tabular rows. $F = 3$ is the number of cross-validation folds, in which training and testing of the discriminator is repeated $F$ times, keeping fixed the generator weights. We use this metric with its corresponding publicly available code (Solatorio & Dupriez, 2023) for a fair comparison with REaLTabFormer (Solatorio & Dupriez, 2023) and SDV (Patki et al., 2016). Specifically, in the conditional generation scenario, we follow (Solatorio & Dupriez, 2023) and we evaluate the *LD-SR* for the "merged" task by concatenating $\boldsymbol{u}$ with all the rows $\boldsymbol{r}$ extracted from a generated time series $\boldsymbol{x}$. The Random Forest is then trained and tested on this "augmented" rows. In case of *MLD-TS*, we concatenate $\boldsymbol{u}$ with the fixed-dimension feature vector extracted from $\boldsymbol{x}$ using the feature extraction library (Christ et al., 2018) (Sec. 5.1), and we use the "augmented" feature vector to train and test CatBoost.

Finally, we introduce an additional metric based on the Machine Learning Efficiency (MLE) (Zhang et al., 2024; Kotelnikov et al., 2023), which is based on the accuracy of a classifier trained on generated data and evaluated on a real data testing set. MLE can also be used to simulate an application scenario in which, for instance, the generated data are used to replace real data (e.g., because protected by privacy or legal constraints, see Sec. 1) in training and testing public machine learning methods. In this case, the classifier accuracy, when trained with synthetic data (only) is usually upper bounded by the accuracy of the same classifier trained on the real data. Similarly to *MLD-TS*, we adapt this metric (which we refer to as *MLE-TS*) to our time series domain using CatBoost as the classifier, fed using a fixed-size feature vector extracted from a given time series (Christ et al., 2018) (Sec. 5.1). For simplicity, we always use binary classification tasks, whose lower bound is $50\%$ (chance level), and in App. D.1 we show how these tasks can be formulated selecting specific attributes of the parent table to be used as target labels.

## D  ADDITIONAL EXPERIMENTS

### D.1  MACHINE LEARNING EFFICIENCY

In this section, we show additional experiments using the *MLE-TS* metric introduced in App. C. Since this metric is based on training a classifier to predict a target label, we can use *MLE-TS* only to evaluate conditional generations, where these labels can be extracted from the parent table of the corresponding dataset (individual time series or rows are not labeled). Moreover, the "merged" task cannot be used in this case because, as defined in (Solatorio & Dupriez, 2023), at inference time, it involves the concatenation of the generated data with the conditioning parent table (App. C), the latter used to extract the ground truth target labels. Specifically:

- In Rossmann, we use the binary attribute "Promo2", indicating the presence of a promo in that store.
- In Airbnb, we use the attribute "n_sessions" which indicates the number of sessions opened by a user. We binarize this attribute predicting whether the user has opened more than 20 sessions ($n\_sessions >= 20$).
- In Age2, we use the attribute "age", indicating the age of a customer. In particular, we predict whether the customer is over 30 years old ($age > 30$).
- In PKDD'99 Financial, we use the attribute "region" which indicates the region a customer belongs to. We predict whether the customer is located in Moravia or in Bohemia (including Prague).

In Tab. 6, "Original" indicates that the classifier has been trained on *real* data and tested on real data, and it is an ideal value of the expected accuracy when the same classifier is trained on synthetic data. In the same table, "child" and "child gt-cond" refer to the tasks used in Tab. 4. Specifically, in both

cases the classifier is trained with the *generated* data and tested on real data. However, in case of "child gt-cond", we use the *real* parent table row values as the classification target labels (see above). Conversely, in case of "child", we use the *generated* parent table rows. This is because, in a realistic scenario, the "child" task corresponds to a situation in which the parent table data are missing and they are generated as well (Sec. 5.3), thus we coherently use the synthetic parent table values to label the corresponding generated time series. The real time series are always labeled with their corresponding real parent table values.

The results in Tab. 6 confirm those reported in Tab. 4, showing that TabDiT significantly outperforms all the other baselines in all the datasets. Specifically, in Age2, TabDiT even surpasses the ideal "Original" accuracy (61.57 versus 60.07), being very close to the ideal case in all the other datasets. We believe that these results show the effectiveness of the synthetic time series generated by our method, which can potentially be used to replace real data in machine learning tasks when, e.g., the real data cannot be made public.

Table 6: Conditional generation results using the *MLE-TS* $\uparrow$ metric (mean and standard deviation over three runs).

| Method | Task | Rossmann | Airbnb | Age2 | PKDD'99 Financial |
|---|---|---|---|---|---|
| Original | - | $66.70_{\pm 8.90}$ | $100.00_{\pm 0.00}$ | $60.07_{\pm 1.74}$ | $61.67_{\pm 2.95}$ |
| SDV | child | $54.10_{\pm 2.60}$ | $93.97_{\pm 1.37}$ | $54.43_{\pm 1.95}$ | $61.30_{\pm 4.81}$ |
| REaLTabFormer | child gt-cond | $53.33_{\pm 2.25}$ | $60.87_{\pm 3.10}$ | $54.53_{\pm 1.42}$ | $61.10_{\pm 2.44}$ |
| | child | $53.37_{\pm 5.88}$ | $61.97_{\pm 1.57}$ | $54.97_{\pm 1.80}$ | $61.47_{\pm 8.29}$ |
| AR Baseline | child gt-cond | $59.27_{\pm 1.27}$ | $100.00_{\pm 0.00}$ | $58.23_{\pm 1.96}$ | $\mathbf{62.03}_{\pm 5.41}$ |
| (ours) | child | $50.37_{\pm 9.26}$ | $59.90_{\pm 2.46}$ | $55.77_{\pm 4.11}$ | $60.73_{\pm 3.54}$ |
| TabDiT (ours) | child gt-cond | $\mathbf{64.43}_{\pm 10.19}$ | $\mathbf{100.00}_{\pm 0.00}$ | $\mathbf{61.57}_{\pm 3.05}$ | $61.30_{\pm 1.76}$ |
| | child | $\mathbf{65.93}_{\pm 9.22}$ | $\mathbf{99.63}_{\pm 0.40}$ | $\mathbf{61.23}_{\pm 2.55}$ | $\mathbf{62.60}_{\pm 3.82}$ |

## D.2 LARGER SCALE EXPERIMENTS

In this section, we use a much larger dataset to show the potentialities of our method to be scaled. Since, as far as we know, heterogeneous time series datasets considerably larger than those used in Sec. 5 are not publicly available, we used a private dataset which we call *Large Scale Bank Data*, and which was provided by an international bank[2]. Large Scale Bank Data consists of several hundred million real bank account transactions of private customers. From this dataset, we randomly selected 100K client bank accounts, corresponding to approximately 87.3M transactions (i.e., rows), which we use to train TabDiT and the AR Baseline. Moreover, we selected another set of 10K bank accounts (not included in the training set) for evaluation. Furthermore, with Large Scale Bank Data we use longer time series, setting $\tau_{max} = 100$, which corresponds to an average of one month of bank transactions of a given customer (see App. E for more details).

In Tab. 7, we compare TabDiT with AR Baseline using the "child gt-cond" task. Note that we were not able to use REaLTabFormer on this large-scale datasets for computational reasons. Indeed the long length of the time series ($\tau_{max} = 100$), combined with the larger number of time series fields (9, as reported in Tab. 9), led to memory usage and time complexity problems during training with REaLTabFormer. Conversely, both the hierarchical architecture of AR Baseline (Sec. 5.2) and our LDM approach (Sec. 1) allow a *much faster and lower memory consumption* training, which made possible to use a huge dataset like Large Scale Bank Data.

In Tab. 7 we report the *MLD-TS*, the *LD-SR*, and the *MLE-TS* metric values. These results show that TabDiT significantly outperforms AR Baseline. Moreover, even when using a large-scale dataset, TabDiT approaches the lower bound of $50\%$ *MLD-TS* accuracy, achieves a score higher than $80\%$ *LD-SR*, and is able to approach the upper bound of *MLE-TS* obtained using the real data.

---

[2]For both privacy and commercial reasons, this dataset cannot be released.

Table 7: Large scale conditional experiments using the "child gt-cond" task and Large Scale Bank Data.

| Method | MLD-TS ↓ | LD-SR ↑ | MLE-TS ↑ |
|---|---|---|---|
| Original | - | - | 73.22 |
| AR Baseline (ours) | 58.03 | 83.33 | 70.41 |
| TabDiT (ours) | **56.77** | **83.72** | **71.38** |

Table 8: PKDD'99 Financial dataset. Analysis of the influence of the CFG hyperparameter values using the **MLD-TS ↓** metric (mean and standard deviation over three runs).

| $p_d$ | $s = 1$ | $s = 2$ | $s = 3$ | $s = 4$ | $s = 5$ |
|---|---|---|---|---|---|
| 0.001 | 83.33 $_{\pm 3.61}$ | 72.67 $_{\pm 3.52}$ | 78.33 $_{\pm 7.86}$ | 71.13 $_{\pm 9.38}$ | 69.33 $_{\pm 7.92}$ |
| 0.005 | 80.47 $_{\pm 8.02}$ | 73.00 $_{\pm 8.84}$ | 72.47 $_{\pm 14.95}$ | **59.50** $_{\pm 10.53}$ | 75.00 $_{\pm 12.30}$ |
| 0.010 | 77.87 $_{\pm 10.40}$ | 70.80 $_{\pm 10.83}$ | 69.50 $_{\pm 13.95}$ | 76.83 $_{\pm 6.40}$ | 73.97 $_{\pm 9.51}$ |
| 0.100 | 80.67 $_{\pm 4.74}$ | 71.10 $_{\pm 19.11}$ | 69.13 $_{\pm 12.90}$ | 66.77 $_{\pm 12.79}$ | 69.63 $_{\pm 16.61}$ |

### D.3 ADDITIONAL ABLATIONS

In this section we present additional ablation studies. In Tab. 8, we use the "child gt-cond" task to investigate the influence of the CFG and its hyperparameter values $s$ and $p_d$ (Sec. 4). For these experiments we use *PKDD'99 Financial* dataset instead of *Age2* (used in all other ablations) because, on *Age2*, TabDiT achieves a nearly ideal *MLD-TS* score ($\sim 50\%$) *even without CFG*, so there is no room for improvement. In Tab. 8, the scale value $s = 1$ corresponds to non CFG (see Eq. (2)), and the results reported in the table clearly shows that CFG is beneficial for conditional generation of tabular data time series, a finding which is aligned with the empirical importance of CFG in the image generation literature (Peebles & Xie, 2023). Using these results, we selected the values $p_d = 0.005$ and $s = 4$ and we used these hyperparameter values *in all the datasets and conditional tasks*. Although a dataset-dependent hyperparameter tuning may likely lead to even better results, we opted for a simpler and computationally less expensive solution based on dataset-agnostic CFG hyperparameters.

For training our VAE we adopt the scheduling proposed in (Zhang et al., 2024), where the reconstruction loss and the KL-divergence loss are balanced using a coefficient $\beta$ which weights the importance of the latter. The value of $\beta$ starts from an initial $\beta_{max}$ and, during training, it is progressively and adaptively reduced. We refer to (Zhang et al., 2024) for more details. In Fig. 2 we use *Age2* to show the impact of $\beta_{max}$, which is evaluated jointly with the number of diffusion steps $T$ of the denoising network (Sec. 3). We evaluate the values of $\beta_{max}$ and $T$ jointly because they are strongly related to each other. This ablation shows that using a relatively small number of diffusion steps (e.g., greater than 100) is sufficient to get good results, and that there is no significant improvement with very long trajectories. Conversely, a higher value of $\beta_{max}$, corresponding to a higher regularization of the latent space, improves the results. In all the datasets and tasks, we use 200 diffusion steps and $\beta_{max} = 5$.

## E DATASETS

In Tab. 9 we report the main characteristics of the datasets used in our experiments. We use six real-world public datasets: *Age2*[3], *Age1*[4], *Leaving*[5], the *PKDD'99 Financial* dataset[6], the *Rossmann* store sales dataset[7] and the *Airbnb* new user bookings dataset[8]. The first three datasets have been

---

[3] Age2 dataset (Fursov et al., 2021)

[4] Age1 dataset (Fursov et al., 2021)

[5] Leaving dataset (Fursov et al., 2021)

[6] PKDD'99 Financial dataset (Berka, 1999)

[7] Rossmann store sales dataset (Solatorio & Dupriez, 2023)

[8] Airbnb new user bookings dataset (Solatorio & Dupriez, 2023)

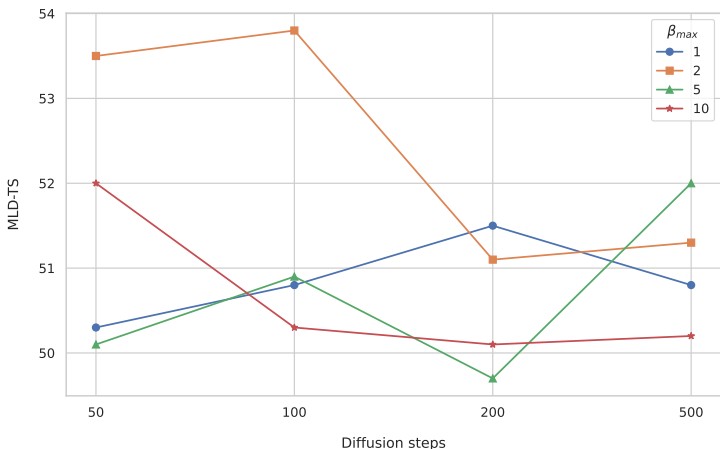

Figure 2: Age2 dataset. Analysis of the influence of the number of diffusion time steps $T$ jointly with the $\beta_{max}$ value using the **MLD-TS** $\downarrow$ metric.

Table 9: Dataset statistics.

| | Age2 | Age1 | Leaving | PKDD'99 Financial | Rossmann | Airbnb | Large Scale Bank Data |
|---|---|---|---|---|---|---|---|
| Total dataset rows | 3,652,757 | 44,117,905 | 490,513 | 1,056,320 | 68,015 | 192,596 | 96,040,010 |
| Total time series | 43,289 | 50,000 | 5,000 | 4,500 | 1,115 | 10,000 | 110,000 |
| Training samples | 38,961 | 45,000 | 4,000 | 3,600 | 892 | 8,000 | 100,000 |
| Testing samples | 4,328 | 5,000 | 1,000 | 900 | 223 | 2,000 | 10,000 |
| $\tau_{max}$ | 50 | 50 | 50 | 50 | 61 | 50 | 100 |
| Time series fields | 3 | 3 | 6 | 6 | 8 | 5 | 9 |
| Time series numerical fields | 1 | 1 | 1 | 2 | 2 | 1 | 1 |
| Time series categorical fields | 1 | 2 | 4 | 3 | 5 | 4 | 7 |
| Time series date/time fields | 1 | 0 | 1 | 1 | 1 | 0 | 1 |
| Parent fields | 4 | 0 | 0 | 5 | 9 | 16 | 8 |
| Parent numerical fields | 2 | 0 | 0 | 0 | 1 | 2 | 0 |
| Parent categorical fields | 2 | 0 | 0 | 4 | 8 | 11 | 8 |
| Parent date/time fields | 0 | 0 | 0 | 1 | 0 | 3 | 0 |

previously used in Fursov et al. (2021), while the last two datasets have been used in Solatorio & Dupriez (2023). *Age2*, *PKDD'99 Financial*, *Rossmann* and *Airbnb* include both parent and time series data, and they can be used for the conditional generation tasks. Conversely, *Age1* and *Leaving* do not include the parent table, thus we used them only for the unconditional generation task. Original source, copyright, and license information are available in the links in the footnotes.

In *Rossmann* and *Airbnb*, we use the same training and testing splits created in (Solatorio & Dupriez, 2023). Specifically, in *Rossmann*, we use 80% of the store data and their associated sales records for training the generator. We use the remaining stores as the testing data (see App. C for more details on how testing data are split for training the discriminators). Again following (Solatorio & Dupriez, 2023), we limit the data used in the experiments from the years 2015-2016 onwards, spanning 2 months of sales data per store. Moreover, in the *Airbnb* dataset, we consider a random sample of 10,000 users for the experiment. We take 8,000 as part of our training data, and we assess the metrics using the 2,000 users in the testing data. We also limit the users considered to those having at most 50 sessions in the data. Regarding *Age2*, *Age1*, *Leaving* and *PKDD'99 Financial*, we use the entire datasets, without any data filtering. The only exception is for the *PKDD'99 Financial* parent table where we used the following fields: district_id, frequency, city, region, and the account creation date.

Tab. 9 also includes the statistics of *Large Scale Bank Data* (App. D.2). In our experiments with this dataset, we used both its parent table and its time series, without any data filtering.

Table 10: Dataset-specific hyperparameter values.

| | | Age2 | Age1 | Leaving | PKDD'99 Financial | Rossmann | Airbnb | Large Scale Bank Data |
|---|---|---|---|---|---|---|---|---|
| TabDiT VAE | Training epochs | 50 | 5 | 100 | 50 | 2,000 | 300 | 3 |
| | Training iterations | 175,950 | 215,100 | 43,200 | 49,150 | 120,000 | 40,800 | 378,800 |
| TabDiT | Training epochs | 150 | 150 | 2,000 | 2,000 | 4,000 | 800 | 180 |
| Denoising network | Training iterations | 45,750 | 52,800 | 64,000 | 58,000 | 28,000 | 50,400 | 118,400 |
| SR-TabDiT VAE | Training epochs | 1,000 | - | - | 5,000 | 6,000 | 20,000 | - |
| | Training iterations | 39,000 | - | - | 20,000 | 6,000 | 180,000 | - |
| SR-TabDiT Denoising | Training epochs | 1,000 | - | - | 3,000 | 6,000 | 3,000 | - |
| network | Training iterations | 170,000 | - | - | 54,000 | 54,000 | 120,000 | - |

Table 11: Dataset-independent hyperparameter values for the VAE.

| Hyperparameter | Value |
|---|---|
| Optimizer | AdamW |
| Learning rate | 5e-05 |
| Training dropout | 0.1 |
| Batch size | 1,024 |
| Model size (parameters) | 2M |
| VAE Encoder Transformer layers | 3 |
| VAE Encoder Transformer heads | 8 |
| VAE Encoder hidden size | 72 |
| VAE Decoder Transformer layers | 3 |
| VAE Decoder Transformer heads | 8 |
| VAE Encoder hidden size | 72 |
| VAE latent space size ($d$) | 792 |
| $\beta_{max}$ | 5 |
| $\beta_{min}$ | 0.05 |
| $\lambda$ | 0.7 |
| $patience$ | 5 |

# F  IMPLEMENTATION DETAILS

In this section, we provide additional implementation details jointly with the values of the hyperparameters used our experiments. Tab. 10 shows the number of training epochs and iterations for both the VAE and the denoising network (TabDiT and SR-TabDiT). Since Large Scale Bank Data was used only for a "child gt-cond" task, we did not train a SR-TabDiT with this dataset.

The model hyperparameter values in Tab. 11 and 12 are shared by all the datasets. Moreover, both TabDiT and SR-TabDiT share the same hyperparameter values, both for the VAE and the denoising network.

Tab. 11 shows the VAE hyperparameters. The encoder $\mathcal{E}_\phi$ and decoder $\mathcal{D}_\varphi$ of the VAE architecture have the same number of layers and the same number of heads, and the same hidden size. The latent space size of the VAE is the same as the DiT-based denoising network $\mathcal{F}_\phi$ hidden size $d$, in order to have the denoising network work directly in the latent space of the VAE model.

The DiT-based denoising network $\mathcal{F}_\phi$ hyperparameters are detailed in Tab. 12. Most of the hyperparameter values are borrowed by the Dit-B model presented in (Peebles & Xie, 2023). We use a standard frequency-based positional embedding, the same as DiT, the only difference being that we have a single dimension input (the time series length) rather than a 2D image. The main hyperparameter of the denoising network that we change is the number of diffusion steps ($T$), which needs to be adapted to our tabular data time series domain (see App. D.3).

Table 12: Dataset-independent hyperparameter values of the denoising network.

| Hyperparameter | Value |
|---|---|
| Optimizer | AdamW |
| Positional encoding | frequency-based |
| Learning rate | 1e-04 |
| Dropout | 0.1 |
| Batch size | 128 |
| Model size (parameters) | 140M |
| DiT depth | 12 |
| DiT num heads | 12 |
| Hidden size ($d$) | 792 |
| Diffusion steps | 200 |
| $p_d$ | 0.005 |
| $s$ | 4 |

Table 13: Dataset-specific total training time (measured in hours).

|  | Age2 | Age1 | Leaving | PKDD'99 Financial | Rossmann | Airbnb | Large Scale Bank Data |
|---|---|---|---|---|---|---|---|
| VAE | 3h | 3h | 1h | 3h | 2h | 2h | 8h |
| Denoising network | 5h | 5h | 5h | 5h | 3h | 3h | 26h |

## G  COMPUTING RESOURCES

All the experiments presented in this paper have been performed on an internal compute node composed of:

- 2 CPUs AMD EPYC 7282 16-Core, for a total of 32 physical and 64 logical cores,

- 256 Gb RAM,

- 4 GPUs Nvidia RTX A6000, each with 48 Gb of memory each, for a total of 192 Gb.

Table 13 shows the training time for the VAE and the denoising network on each dataset.

## H  QUALITATIVE RESULTS

### H.1  NUMERICAL FIELD REPRESENTATIONS

In this section, we show some qualitative results using the numerical representation methods evaluated in Tab. 1 and presented in detail in App. A. Specifically, we use the *Airbnb* dataset and we select the "secs_elapsed" attribute. This is the field with the highest variability, with values ranging from 0 to 1.8M, a mean of 3.3K and a standard deviation of approximately 13K.

In Fig. 3 to 6 we compare to each other the distributions of all evaluated numerical representations. In every plot, the $x$ axis is based on a logarithmic scale and a fixed number of 50 bins for both the real and the generated numerical values. For each bin, in the $y$ axis we show the percentage of tabular rows that contain numerical values that belong to that bin.

In *Quantize* (Fig. 3), the "secs_elapsed" field values are quantized using equal-size bins of 360 seconds (1 hour) and we generate the bin center. As depicted in the figure, this method struggles in representing small numerical values, which are all grouped in a few bins. On the other hand, the *Linear transf* representation (Fig. 4) tends to under-sample the tails of the distribution. In the last two figures, we qualitatively evaluate the *Fixed digit seq* (Fig. 5) and our *variable-range decimal representation* (Fig. 6). The corresponding distributions show that our variable-range representation is more accurate in reproducing numerical values. As mentioned in App. A, we believe that one of

the reasons why the *Fixed digit seq* representation is sub-optimal is that its longer length leads, on average, to a more difficult decoding problem.

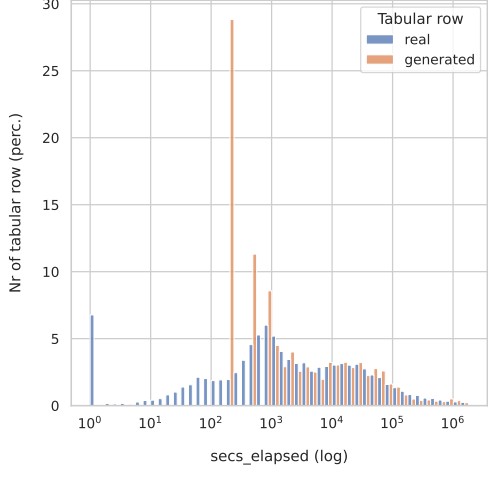

Figure 3: Real and generated distributions of the values of the "secs_elapsed" attribute using *Quantize* as the numerical field value representation.

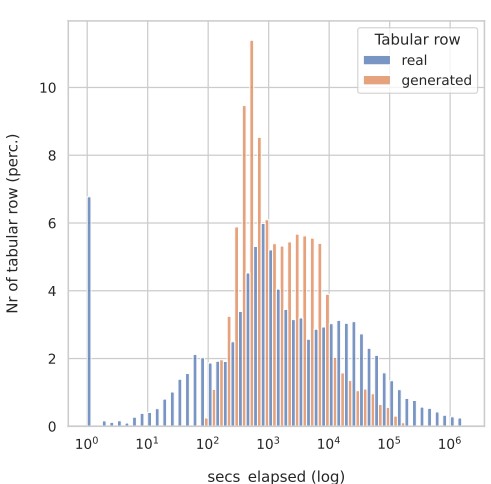

Figure 4: Real and generated distributions of "secs_elapsed" using *Linear transf.*

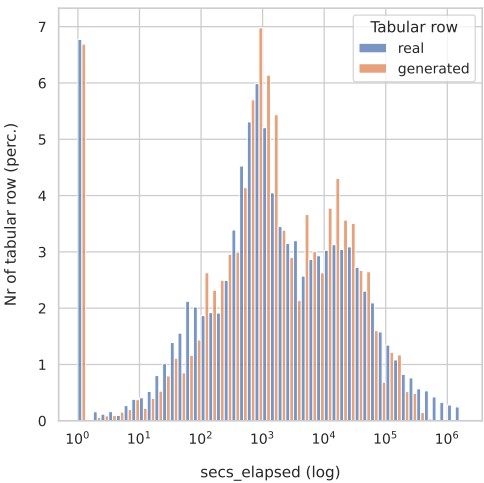

Figure 5: Real and generated distributions of "secs_elapsed" using *Fixed digit seq.*

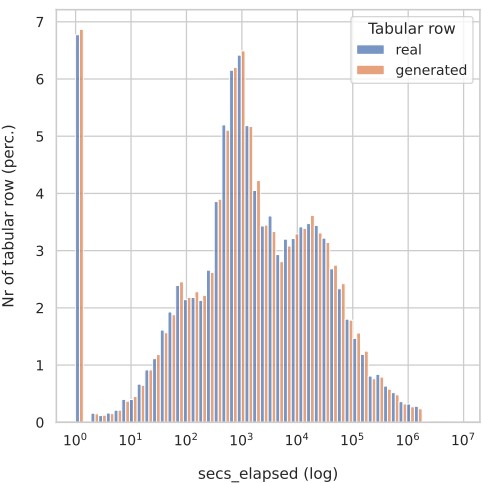

Figure 6: Real and generated distributions of "secs_elapsed" using *variable-range decimal representation*.

## H.2 TIME SERIES LENGTH

In Fig. 7 and 8 we show the distribution of the real and the generated time series lengths on *Airbnb*. In the first plot, we use the *W/o length pred* (Sec. 5.2) method: at inference time the sequence length is randomly sampled using a mono-modal Gaussian distribution fitted on the length of the training time series. On the other hand, in Fig. 8 we use our padding rows (Sec. 4.2) to predict the time series length. Fig. 7 and 8 show that, in the latter case, the time series length distribution is more accurately reproduced.

Fig. 9 and 10 show the distributions of the difference between the real and the generated time series length using a "child gt-cond" task. Specifically, given a real parent table $\boldsymbol{u}$, representing a client of the *Airbnb* dataset, we generate a time series $\hat{\boldsymbol{x}}$ using both our full method (Fig. 10) and *W/o length pred* (Fig. 9). In both cases we compute the difference between the length of the generated series $\hat{\boldsymbol{x}}$ ($\tau_{\hat{\boldsymbol{x}}}$) and the length of the real time series $\boldsymbol{x}$ ($\tau_{\boldsymbol{x}}$) associated with $\boldsymbol{u}$. The shorter the difference, the better the method in predicting the real length. Fig. 9 and 10 show these differences for the two methods. Specifically, in Fig. 9, since $\tau_{\hat{\boldsymbol{x}}}$ is sampled from a mono-modal Gaussian fitted on the training set (Sec. 4.2), it is independent of $\boldsymbol{u}$. As a result, the denoising network cannot predict the correct time series length. On the other hand, Fig. 10 shows that, in case of our full-method, the distribution of the difference between $\tau_{\hat{\boldsymbol{x}}}$ and $\tau_{\boldsymbol{x}}$ is very close to zero, showing the effectiveness of predicting the series length jointly with its content (Sec. 1).

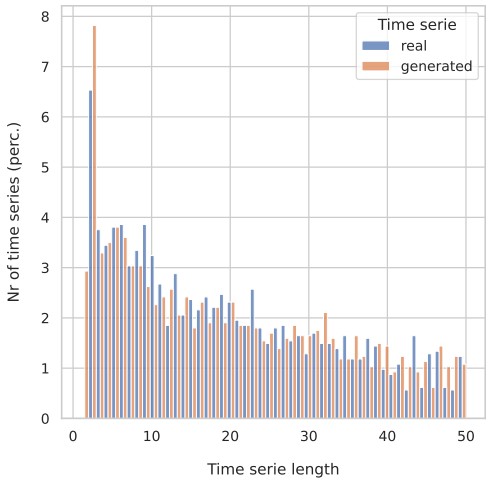

Figure 7: Distribution of the lengths of the real and the generated time series, using *W/o length pred* method.

Figure 8: Distribution of the lengths of the real and the generated time series, using our padding row generation (sec. 4.2).

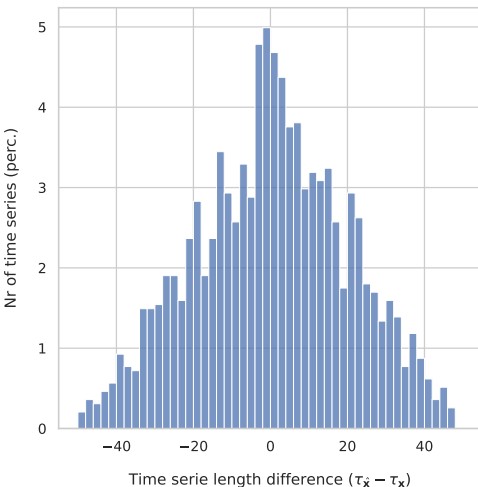

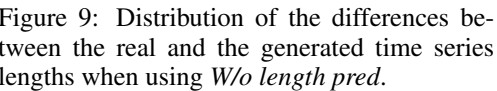

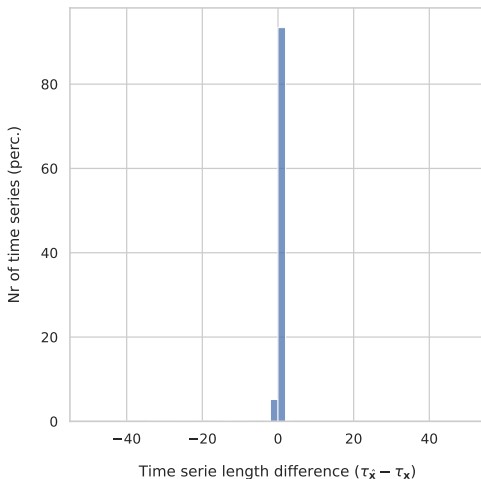

Figure 9: Distribution of the differences between the real and the generated time series lengths when using *W/o length pred*.

Figure 10: Distribution of the differences between the real and the generated time series lengths when using padding row prediction.

### H.3 TIME SERIES EXAMPLES

In this section, we show some examples of real and generated time series to qualitatively evaluate the generation results. We use the "child gt-cond" task on the *PKDD'99 Financial* dataset and on the *Airbnb* dataset. We also use the unconditional generation task on the *Leaving* dataset.

In the conditional task (Fig. 11-Fig. 16), we first provide an example of a real parent table row, used as the conditional ground truth information, then we show the first ten rows of the corresponding real time series, and finally the generated one. In these examples, we compare TabDiT with REaLTabFormer. In the unconditional task, we provide the first ten rows of a real time series example (Fig. 17), a time series example generated using TabDiT (Fig. 18), and a time series example generated using AR baseline (Fig. 19).

The results in Fig. 13 and Fig. 19 show that both REaLTabFormer and the AR baseline make some errors in generating correctly time-ordered dates in the time series. This may be an important error in real-world applications.

**Real parent table row**

| district_id | frequency | city | region | year | month | day |
|---|---|---|---|---|---|---|
| 61 | POPLATEK MESICNE | Trebic | south Moravia | 94 | 6 | 17 |

**Real time series**

| Year | Month | Day | type_trans | operation | k_symbol | amount_trans | balance |
|---|---|---|---|---|---|---|---|
| 96 | 10 | 8 | PRIJEM | VKLAD | None | 13818.0 | 48891.9 |
| 96 | 10 | 12 | VYDAJ | PREVOD NA UCET | UVER | 3757.0 | 45134.9 |
| 96 | 10 | 16 | VYDAJ | VYBER | None | 4440.0 | 40694.9 |
| 96 | 10 | 20 | VYDAJ | VYBER | None | 3600.0 | 37094.9 |
| 96 | 10 | 27 | VYDAJ | VYBER | None | 5280.0 | 31814.9 |
| 96 | 10 | 31 | VYDAJ | VYBER | SLUZBY | 14.6 | 31970.6 |
| 96 | 10 | 31 | PRIJEM | None | UROK | 170.3 | 31985.2 |
| 96 | 11 | 3 | VYDAJ | VYBER | None | 5600.0 | 26370.6 |
| 96 | 11 | 8 | PRIJEM | VKLAD | None | 13818.0 | 40188.6 |
| 96 | 11 | 12 | VYDAJ | PREVOD NA UCET | UVER | 3757.0 | 36431.6 |

Figure 11: *PKDD'99 Financial* dataset: an example of a **real time series** and its corresponding real parent table row.

**Real parent table row**

| district_id | frequency | city | region | year | month | day |
|---|---|---|---|---|---|---|
| 61 | POPLATEK MESICNE | Trebic | south Moravia | 94 | 6 | 17 |

**Generated time series**

| Year | Month | Day | type_trans | operation | k_symbol | amount_trans | balance |
|---|---|---|---|---|---|---|---|
| 96 | 2 | 18 | VYDAJ | VYBER | None | 720.0 | 19090.0 |
| 96 | 2 | 28 | VYDAJ | VYBER | None | 2800.0 | 16690.0 |
| 96 | 2 | 29 | VYDAJ | VYBER | SLUZBY | 14.6 | 16750.0 |
| 96 | 2 | 29 | PRIJEM | None | UROK | 80.5 | 16750.0 |
| 96 | 3 | 5 | VYDAJ | PREVOD NA UCET | | 3146.0 | 14410.0 |
| 96 | 3 | 8 | PRIJEM | VKLAD | None | 6914.0 | 20790.0 |
| 96 | 3 | 12 | VYDAJ | VYBER | None | 1500.0 | 20420.0 |
| 96 | 3 | 13 | PRIJEM | VKLAD | None | 5718.0 | 26530.0 |
| 96 | 3 | 31 | VYDAJ | VYBER | SLUZBY | 14.6 | 26690.0 |
| 96 | 3 | 31 | PRIJEM | None | UROK | 87.5 | 26700.0 |

Figure 12: *PKDD'99 Financial* dataset: an example of a **generated time series** conditioned on a ground truth parent table row using TabDiT.

**Real parent table row**

| district_id | | frequency | city | region | year | month | day |
|---|---|---|---|---|---|---|---|
| | 61 | POPLATEK MESICNE | Trebic | south Moravia | 94 | 6 | 17 |

**Generated time series**

| Year | Month | Day | type_trans | operation | k_symbol | amount_trans | balance |
|---|---|---|---|---|---|---|---|
| 96 | 11 | 7 | VYDAJ | PREVOD NA UCET | | 76.8 | 13675.2 |
| 96 | 12 | 16 | VYDAJ | PREVOD NA UCET | SIPO | 4508.0 | 19801.2 |
| 96 | 12 | 2 | VYDAJ | VYBER | SLUZBY | 1372.0 | 14904.4 |
| 96 | 11 | 14 | PRIJEM | PREVOD Z UCTU | DUCHOD | 180.0 | 16491.9 |
| 97 | 1 | 6 | VYDAJ | PREVOD NA UCET | None | 1823.0 | 23939.3 |
| 96 | 12 | 7 | VYDAJ | VYBER | None | 4527.0 | 15727.0 |
| 96 | 12 | 6 | VYDAJ | VYBER | SLUZBY | 1771.0 | 11381.1 |
| 97 | 4 | 5 | VYDAJ | PREVOD NA UCET | SIPO | 71.6 | 66265.1 |
| 97 | 1 | 11 | VYDAJ | VYBER | None | 114.3 | 19241.6 |
| 96 | 10 | 3 | VYDAJ | PREVOD NA UCET | SIPO | 14.0 | 15248.9 |

Figure 13: *PKDD'99 Financial* dataset: an example of a **generated time series** conditioned on a ground truth parent table row using REaLTabFormer. The sequence of the dates is not chronologically ordered.

**Real parent table row**

| CompetitionDistance | Promo2SinceWeek | CompetitionOpenSinceYear | CompetitionOpenSinceMonth | Promo2SinceYear | StoreType | Assortment | PromoInterval | Promo2 |
|---|---|---|---|---|---|---|---|---|
| 1420.0 | 40.0 | 2012.0 | 10.0 | 2014.0 | a | a | Jan,Apr,Jul,Oct | 1 |

**Real time series**

| Open | Promo | StateHoliday | SchoolHoliday | DayOfWeek | Customers | Sales | Date_month | Date_day |
|---|---|---|---|---|---|---|---|---|
| 1 | 1 | 0 | 0 | 5 | 743.0 | 7509.0 | 7 | 31 |
| 1 | 1 | 0 | 0 | 4 | 687.0 | 7171.0 | 7 | 30 |
| 1 | 1 | 0 | 0 | 3 | 647.0 | 6926.0 | 7 | 29 |
| 1 | 1 | 0 | 0 | 2 | 696.0 | 7432.0 | 7 | 28 |
| 1 | 1 | 0 | 0 | 1 | 753.0 | 8528.0 | 7 | 27 |
| 0 | 0 | 0 | 0 | 7 | 0.0 | 0.0 | 7 | 26 |
| 1 | 0 | 0 | 0 | 6 | 710.0 | 6887.0 | 7 | 25 |
| 1 | 0 | 0 | 0 | 5 | 593.0 | 5056.0 | 7 | 24 |
| 1 | 0 | 0 | 0 | 4 | 586.0 | 5557.0 | 7 | 23 |
| 1 | 0 | 0 | 0 | 3 | 491.0 | 4603.0 | 7 | 22 |

Figure 14: *Airbnb* dataset: an example of a **real time series** and a real parent table row.

**Real parent table row**

| CompetitionDistance | Promo2SinceWeek | CompetitionOpenSinceYear | CompetitionOpenSinceMonth | Promo2SinceYear | StoreType | Assortment | PromoInterval | Promo2 |
|---|---|---|---|---|---|---|---|---|
| 1420.0 | 40.0 | 2012.0 | 10.0 | 2014.0 | a | a | Jan,Apr,Jul,Oct | 1 |

**Generated time series**

| Open | Promo | StateHoliday | SchoolHoliday | DayOfWeek | Customers | Sales | Date_month | Date_day |
|---|---|---|---|---|---|---|---|---|
| 1 | 1 | 0 | 0 | 5 | 593.0 | 6882.0 | 7 | 31 |
| 1 | 1 | 0 | 0 | 4 | 585.0 | 6349.0 | 7 | 30 |
| 1 | 1 | 0 | 0 | 3 | 505.0 | 5094.0 | 7 | 29 |
| 1 | 1 | 0 | 0 | 2 | 571.0 | 6752.0 | 7 | 28 |
| 1 | 1 | 0 | 0 | 1 | 596.0 | 7502.0 | 7 | 27 |
| 0 | 0 | 0 | 0 | 7 | 0.0 | 0.0 | 7 | 26 |
| 1 | 0 | 0 | 0 | 6 | 471.0 | 5136.0 | 7 | 25 |
| 1 | 0 | 0 | 0 | 5 | 462.0 | 4352.0 | 7 | 24 |
| 1 | 0 | 0 | 0 | 4 | 470.0 | 4249.0 | 7 | 23 |
| 1 | 0 | 0 | 0 | 3 | 371.0 | 3722.0 | 7 | 22 |

Figure 15: *Airbnb* dataset: an example of a **generated time series** conditioned on a ground truth parent table row using TabDiT.

**Real parent table row**

| CompetitionDistance | Promo2SinceWeek | CompetitionOpenSinceYear | CompetitionOpenSinceMonth | Promo2SinceYear | StoreType | Assortment | PromoInterval | Promo2 |
|---|---|---|---|---|---|---|---|---|
| 1420.0 | 40.0 | 2012.0 | 10.0 | 2014.0 | a | a | Jan,Apr,Jul,Oct | 1 |

**Generated time series**

| Open | Promo | StateHoliday | SchoolHoliday | DayOfWeek | Customers | Sales | Date_month | Date_day |
|---|---|---|---|---|---|---|---|---|
| 1 | 1 | 0 | 1 | 5 | 732 | 10200 | 7 | 31 |
| 1 | 1 | 0 | 1 | 4 | 494 | 4633 | 7 | 30 |
| 1 | 1 | 0 | 1 | 3 | 595 | 8110 | 7 | 29 |
| 1 | 1 | 0 | 1 | 2 | 791 | 5896 | 7 | 28 |
| 1 | 1 | 0 | 1 | 1 | 1134 | 7871 | 7 | 27 |
| 0 | 0 | 0 | 0 | 7 | 0 | 0 | 7 | 26 |
| 1 | 0 | 0 | 0 | 6 | 407 | 5000 | 7 | 25 |
| 1 | 0 | 0 | 1 | 5 | 785 | 5642 | 7 | 24 |
| 1 | 0 | 0 | 1 | 4 | 984 | 4870 | 7 | 23 |
| 1 | 0 | 0 | 1 | 3 | 688 | 4380 | 7 | 22 |

Figure 16: *Airbnb* dataset: an example of a **generated time series** conditioned on a ground truth parent table row using REaLTabFormer.

| year | month | day | hour | minute | second | channel_type | trx_category | currency | MCC | amount |
|---|---|---|---|---|---|---|---|---|---|---|
| 2017 | 7 | 10 | 0 | 0 | 0 | type1 | | POS | 810 | 5211 | 1471.49 |
| 2017 | 7 | 10 | 4 | 59 | 42 | type1 | | POS | 810 | 5999 | 1300.00 |
| 2017 | 7 | 11 | 0 | 0 | 0 | type1 | | POS | 810 | 5411 | 102.00 |
| 2017 | 7 | 12 | 0 | 0 | 0 | type1 | | POS | 810 | 5411 | 312.00 |
| 2017 | 7 | 12 | 0 | 0 | 0 | type1 | | POS | 810 | 5211 | 1822.63 |
| 2017 | 8 | 1 | 0 | 0 | 0 | type1 | | POS | 810 | 5411 | 184.00 |
| 2017 | 8 | 11 | 0 | 0 | 0 | type1 | WD_ATM_PARTNER | 810 | 6011 | 7000.00 |
| 2017 | 8 | 11 | 0 | 0 | 0 | type1 | | POS | 810 | 5691 | 4027.00 |
| 2017 | 8 | 11 | 0 | 0 | 0 | type1 | | POS | 810 | 5691 | 499.00 |
| 2017 | 8 | 11 | 0 | 0 | 0 | type1 | | POS | 810 | 7922 | 980.00 |

Figure 17: *Leaving* (unconditional task): an example of a **real time series**.

| year | month | day | hour | minute | second | channel_type | trx_category | currency | MCC | amount |
|---|---|---|---|---|---|---|---|---|---|---|
| 2017 | 4 | 20 | 0 | 0 | 0 | type2 | POS | 810 | 5411 | 756.0 |
| 2017 | 4 | 22 | 0 | 0 | 0 | type2 | POS | 810 | 8999 | 1600.0 |
| 2017 | 4 | 22 | 0 | 0 | 0 | type2 | POS | 810 | 5912 | 500.0 |
| 2017 | 4 | 28 | 0 | 0 | 0 | type2 | POS | 810 | 5977 | 3028.0 |
| 2017 | 5 | 1 | 0 | 0 | 0 | type2 | POS | 810 | 5411 | 749.0 |
| 2017 | 5 | 12 | 14 | 41 | 43 | type2 | WD_ATM_ROS | 810 | 6011 | 1000.0 |
| 2017 | 5 | 21 | 0 | 0 | 0 | type2 | POS | 810 | 5411 | 1068.0 |
| 2017 | 5 | 21 | 19 | 8 | 6 | type2 | WD_ATM_ROS | 810 | 6011 | 2100.0 |
| 2017 | 5 | 22 | 0 | 0 | 0 | type2 | WD_ATM_OTHER | 810 | 6011 | 2400.0 |
| 2017 | 5 | 22 | 0 | 0 | 0 | type2 | POS | 810 | 5411 | 857.1 |

Figure 18: *Leaving* (unconditional task): an example of a **generated time series** using TabDiT.

| year | month | day | hour | minute | second | channel_type | trx_category | currency | MCC | amount |
|---|---|---|---|---|---|---|---|---|---|---|
| 2017 | 3 | 2 | 10 | 4 | 15 | type1 | WD_ATM_ROS | 810 | 6011 | 40000.00 |
| 2017 | 3 | 28 | 0 | 0 | 0 | type1 | POS | 810 | 5331 | 17660.00 |
| 2017 | 4 | 26 | 0 | 0 | 0 | type1 | POS | 810 | 5964 | 33640.00 |
| 2017 | 4 | 8 | 0 | 0 | 0 | type1 | POS | 810 | 5411 | 3775.00 |
| 2017 | 4 | 9 | 0 | 41 | 36 | type1 | POS | 810 | 5814 | 490.00 |
| 2017 | 5 | 3 | 0 | 0 | 0 | type1 | POS | 810 | 5942 | 102.00 |
| 2017 | 5 | 22 | 0 | 0 | 0 | type1 | POS | 810 | 5411 | 4998.00 |
| 2017 | 5 | 18 | 0 | 0 | 0 | type1 | POS | 810 | 5681 | 3240.00 |
| 2017 | 5 | 18 | 10 | 23 | 6 | type1 | WD_ATM_ROS | 810 | 5912 | 4564.00 |
| 2017 | 6 | 15 | 12 | 29 | 1 | type1 | WD_ATM_ROS | 810 | 3381 | 0.35 |

Figure 19: *Leaving* (unconditional task): an example of a **generated time series** using AR baseline. The sequence of the dates is not chronologically ordered.

