# OpenReview forum: "Diffusion Transformers for Tabular Data Time Series Generation"
_ICLR.cc/2025/Conference — ICLR 2025 Poster_

### Official Review · Reviewer_PrJn · 2024-10-30

**Soundness:** 2
**Presentation:** 1
**Contribution:** 3
**Rating:** 5
**Confidence:** 2

**Summary:**

This work introduces a Diffusion Transformers (DiTs) based approach for generating time series of tabular data, addressing the challenges of data heterogeneity and variable sequence lengths. Inspired by the success of DiTs in image and video generation, the framework is extended for tabular data series. Extensive experiments on six datasets demonstrate that the proposed method significantly outperforms previous approaches, with the code to be made public upon acceptance of the article.

**Strengths:**

The paper introduces a novel Diffusion Transformers (DiT)-like approach combined with an Auto-Regressive (AR) Variational Autoencoder (VAE) decoder and explicit padding prediction mechanisms. This design effectively addresses the complex challenges associated with generating time series of tabular data, such as maintaining intra-row statistical dependencies and managing variable-length sequences. This comprehensive framework ensures that the generated data preserves the intricate relationships within each row and adapts to varying sequence lengths, which are critical for accurate and realistic data generation.

The authors propose a new metric specifically tailored for the evaluation of tabular data time series generation. This metric provides a standardized and objective means to assess the quality and performance of generated data, facilitating more reliable comparisons across different methods. By introducing this metric, the paper sets a benchmark for future research in this domain, promoting the development of more robust and effective data generation techniques.

The proposed method is rigorously evaluated using a diverse set of public datasets, demonstrating its versatility and robustness across various types of tabular data. This extensive experimental validation highlights the method's superior performance compared to existing approaches, showcasing its applicability to a wide range of real-world scenarios. The thorough evaluation not only underscores the effectiveness of the approach but also provides a solid foundation for its adoption and further development in the field.

**Weaknesses:**

While the paper introduces numerous model modifications aimed at addressing the challenges of tabular data time series generation, it lacks ablation experiments to isolate and evaluate the impact of each modification. Ablation studies are crucial for understanding the contribution of individual components to the overall performance of the model. Without these experiments, it is difficult to determine which modifications are most effective and how they interact to improve the model's capabilities.

The paper does not include any case studies that showcase the generated tabular data time series. Given that the primary focus of the research is on generating realistic and useful time series data, it is surprising that the paper does not provide examples to illustrate the differences between the outputs of various models. Including case studies would provide valuable insights into the practical implications of the proposed method and allow readers to visually assess the quality and realism of the generated data.

The paper does not present experiments that demonstrate the practical utility of the generated time series tabular data. It is important to show how the generated data can be used in real-world applications, such as improving predictive modeling, anomaly detection, or decision-making processes. Without such experiments, it is challenging to gauge the practical relevance and impact of the proposed method. Demonstrating the usefulness of the generated data in specific applications would significantly strengthen the paper's contributions and highlight its value to practitioners.

**Questions:**

See weaknesses section

---

### Official Review · Reviewer_61wY · 2024-10-30

**Soundness:** 3
**Presentation:** 3
**Contribution:** 3
**Rating:** 6
**Confidence:** 3

**Summary:**

The paper introduces a novel method for generating time series of tabular data using DiTs. The authors highlight the challenges in this domain, such as the heterogeneity of tabular data and the variable length of time series. They propose the TabDiT framework, which extends the LDM paradigm to handle the complexities of tabular data. The method involves decomposing the representation of time series into independent tabular rows compressed by a VAE and combined by a Transformer-based denoising network. Through extensive experiments on six datasets, the paper demonstrates the superiority of TabDiT over existing methods in both conditional and unconditional generation tasks.

**Strengths:**

- Utilizing DiT for tabular data time series generation seems an innovative contribution to the field, which extends DiT to multi-modal data distributions that could pave the way for future research.

- The authors conduct extensive experiments on six diverse datasets, covering a wide range of application scenarios. This includes both conditional and unconditional generation tasks, providing a thorough assessment of the method's performance. The ablation study also strengthens the credibility of the paper results.

- The proposed method is methodologically sound, with detailed explanations of the VAE encoder/decoder, the denoising network, and the variable-length sequence generation process. The introduction of a variable-range decimal representation for numerical values and an autoregressive VAE decoder are novel and improve the quality and consistency of the generated data.

**Weaknesses:**

- While the paper introduces new evaluation metrics for tabular data time series generation, these metrics may not be universally accepted or standardized. The authors should provide a more in-depth comparison with existing metrics and justify their choices.

- Some parts of the methodology, particularly the training process and parameter settings, could benefit from more detailed explanations. Providing clear guidelines and examples would help in better understanding and implementing the proposed approach.

**Questions:**

- How does the proposed method scale with increasing dataset size and complexity? Have you tested the method on larger datasets, and if so, what were the results?

- The paper includes several hyperparameters, such as the number of digits for numerical representation and the length of the latent sequence. How sensitive is the performance to these hyperparameters, and how were they chosen?

---

### Official Review · Reviewer_Eh8R · 2024-11-04

**Soundness:** 3
**Presentation:** 3
**Contribution:** 3
**Rating:** 6
**Confidence:** 4

**Summary:**

This paper proposes a system based on Diffusion Transformer to generate tabular data series. Particularly the difficulties of handling heterogeneity and variable length of tabular data are tackled. The paper makes a couple of proposals such as 1) transformer denoising combined with latent encoding of VAE, 2) Autoregressive VAE decoder, 3) several data encoding techniques. Experimental results are presented with multiple datasets using proper metrics for the tasks.

**Strengths:**

Each proposal is clearly described and well motivated to solve the targeted issue. Experiments show that the proposed system outperforms the baseline methods in a large margin across different datasets for both unconditional and conditional generation tasks.

**Weaknesses:**

There is a minor question around VAE encoder/decoder which are listed in Questions below.

**Questions:**

- In VAE decoder, one naïve architecture is to feed z instead of [Start] without using the attention from z. Would it work at all, and is there ablation study to compare this and the proposed approach?

---

### Meta-Review · Area_Chair_BKZe · 2024-12-26

**Metareview:**

This paper introduces TabDiT, a framework based on Diffusion Transformers (DiTs) for generating time-series tabular data. Specifically, the paper learns a DiT in the latent space of VAE over the tabular data, where an autoregressive decoder is used to produce data in variable lengths. Additional encoding innovations like variable-range decimal representation for numerical data and explicit padding prediction are proposed to improve modeling accuracy. The authors also propose a new metric for tabular time-series data generation.

**Strengths** (1) Extends diffusion models to tabular time-series data, addressing unique challenges like heterogeneity and sequence variability. (2) Evaluations span six datasets, showcasing the versatility of the method. (3) Proposes a new evaluation metric for tabular time-series data generation

**Weaknesses** (1) The original version of the paper lacks necessary examples and visualizations, which are later addressed in the updated version; (2) The reviewer raised the question about a discussion of how well the method scales with larger datasets or more complex tabular structures;

**Decision** While the paper has received borderline reviews, all the reviewers agreed on the importance of the investigated problem of generating structured data using diffusion models. By checking the authors' responses, most of the reviewers' concerns were addressed with clear evidence. Therefore, I am leaning toward an acceptance.

**Additional Comments On Reviewer Discussion:**

None of the reviewers engaged with the authors during the discussion phase, which makes the decision-making solely based on the original reviews and the authors' rebuttal.

---

### Decision · Program_Chairs · 2025-01-22

Accept (Poster)